# Learning-To-Measure: In-Context Active Feature Acquisition

**Yuta Kobayashi** [* 1]  **Zilin Jing** [1]  **Jiayu Yao** [1]  **Hongseok Namkoong** [1]  **Shalmali Joshi** [1]

## Abstract

Active feature acquisition (AFA) is a sequential decision-making problem where the goal is to improve model performance for test instances by adaptively selecting which features to acquire. In practice, AFA methods often learn from retrospective data with systematic missingness in the features and limited task-specific labels. To address this limitation, we introduce Learning-to-Measure (L2M), a meta-learning framework that consists of i) reliable uncertainty quantification over unseen tasks, and ii) an uncertainty-guided feature acquisition agent that maximizes conditional mutual information. We demonstrate an autoregressive pre-training approach that underpins reliable uncertainty quantification and feature acquisition across tasks with arbitrary missingness. L2M operates directly on datasets with retrospective missingness and performs the task in-context, eliminating per-task retraining. Across synthetic and real-world tabular benchmarks, L2M matches or surpasses task-specific baselines, particularly under scarce labels and high missingness rates.

## 1. Introduction

Machine learning (ML) methods typically operate under the assumption that all input features are available at inference time. However, this assumption does not hold in scenarios where acquiring certain features involves significant costs or risks, such as medical diagnostics (Erion et al., 2022). For example, acquiring imaging data or invasive biopsies may incur substantial financial costs and pose potential risks to patient safety (Callender et al., 2021). In such cases, it becomes necessary to weigh the expected predictive benefit of each feature against its acquisition cost.

Active feature acquisition (AFA) addresses this problem by learning an agent to adaptively select which features to acquire or observe for each sample (Ma et al., 2018; Shim et al., 2018; von Kleist et al., 2023). AFA is naturally a sequential decision-making problem, where past feature acquisitions informs future acquisition decisions. Prior AFA work uses either greedy acquisition strategies that maximize estimates of one-step expected information gain (Ma et al., 2018; Gong et al., 2019; Covert et al., 2023; Chattopadhyay et al., 2023; Gadgil et al., 2024), or reinforcement learning approaches that learn value (or Q-) functions for multi-step feature acquisition (Shim et al., 2018; Kachuee et al., 2019; Janisch et al., 2019; Li & Oliva, 2021).

Most AFA methods suffer from several key limitations. First, historical training data often exhibits retrospective missingness, defined as the absence of features arising from prior data collection policies such as clinical protocols, resource constraints, and workflow decisions, and patient behavior. For example, in chest-pain triage, clinical guidelines prioritize first-line laboratory tests before ordering imaging or invasive studies, so missingness in the historical record depends directly on prior observations. The literature shows that methods which do not explicitly account for the missingness structure tend to inherit acquisition bias and produce poorly calibrated uncertainty estimates (von Kleist et al., 2025; 2023). Common remedies are insufficient: feature imputation methods ignore feature informativeness, while models that ignore missingness without properly calibrating uncertainty tend to reproduce existing missingness patterns.

Second, existing AFA methods are designed for single, predetermined tasks rather than general-purpose capability aligned with the foundation model paradigm (Bommasani et al., 2021). Moreover, many prior approaches rely on complex latent-variable models with heuristic approximations to make generative modeling feasible. These methods typically obtain uncertainty through posterior sampling, which is often difficult to scale and unreliable, especially in high-dimensional settings (Ma et al., 2018; Li & Oliva, 2021; Peis et al., 2022).

To address these challenges, we introduce Learning-to-Measure (**L2M**), an in-context AFA approach that leverages the uncertainty quantification capabilities of pre-trained sequence models (Nguyen & Grover, 2022; Ye & Namkoong, 2024; Mittal et al., 2026). At its core, **L2M** couples princi-

---

[1]Columbia University, NY, U.S.A. Correspondence to: Yuta Kobayashi, Shalmali Joshi <yk3043@cumc.columbia.edu, shalmali.joshi@columbia.edu>.

*Proceedings of the 43rd International Conference on Machine Learning*, Seoul, South Korea. PMLR 306, 2026. Copyright 2026 by the author(s).

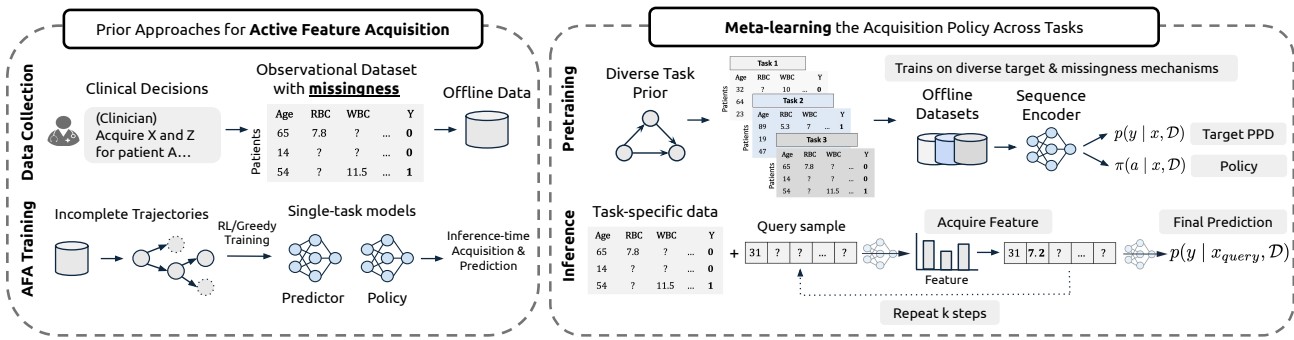

*Figure 1.* Overview: (*Left*) Prior AFA approaches train single-task models using greedy or RL methods on a fixed observational dataset, which may contain missing features due to data collection practices. To handle the incomplete trajectories that result from this retrospective missingness, these methods often rely on deterministic imputations. (*Right*) Our approach pretrains a single sequence model with a target head and a policy head on a diverse task prior from which training datasets are sampled. The target head outputs a posterior predictive distribution, and the policy head selects features that minimize uncertainty in this distribution. By exposing the model to diverse target functions and missingness mechanisms during pretraining, the task prior yields calibrated uncertainty estimates across varying dataset sizes and levels of feature missingness.

pled uncertainty estimation with a feature acquisition policy. **L2M** operates directly on retrospectively missing data and solves the AFA task in-context. Under standard assumptions such as missing at random (MAR) and sufficient coverage of acquisition patterns, **L2M** can be applied without task-specific retraining.

**L2M** consists of two stages: (i) pretraining across tasks with diverse missingness mechanisms to quantify predictive uncertainty of a target variable given partially observed inputs, and (ii) meta-training a policy network to acquire features that reduce this predictive uncertainty. We implement the first stage using sequence modeling over data sequences to capture reliable beliefs under missingness. In the second stage, we optimize a policy end-to-end using a smooth, differentiable approximation of information gain. **L2M** removes the need for latent-variable approximations, performs calibrated and scalable uncertainty estimation via direct sequence prediction. This yields a principled approach to sequential information acquisition across tasks. Figure 1 depicts the schematic of the **L2M** framework at inference.

Our contributions are the following:

1. **Meta-learning AFA across diverse tasks and missingness patterns:** We formalize the problem of meta-learning AFA policies across time-invariant tasks, where the feature values do not change over time, with diverse data distributions and retrospective missingness patterns.

2. **Combining uncertainty estimation and decision-making via sequence modeling:** We propose **L2M**, a scalable transformer-based approach for end-to-end sequential information maximization. The sequence model provides reliable uncertainty estimates for a target variable given partially observed inputs. We show how to leverage these estimates to acquire features that minimize expected loss under

both greedy and lookahead acquisition strategies. To learn the acquisition policy, we design a smooth, differentiable approximation of the optimization objective, yielding a fully auto-differentiable training framework. To our knowledge, this is the first work to demonstrate that such a formulation enables meta-learning of both greedy and planning-based AFA policies that generalize across diverse tasks.

3. **Robustness to limited labeled data and missingness:** We empirically show that our meta-learning-based approach, **L2M**, outperforms task-specific baselines across datasets of varying sizes and degrees of missingness, particularly when labeled data are scarce, and feature missingness is high.

We first formalize the meta-AFA problem in Section 3. We then extend meta-AFA to settings with missing data and establish identifiability conditions under which the resulting optimization problem can be solved using observational data (Section 3.1). Section 4 presents the core components of **L2M** and our pre-training procedure. In Section 4.1, we introduce the sequence modeling–based meta-learning framework for predicting the outcome. Section 4.2 outlines our proposed policy optimization objective. Section 5 demonstrates the empirical utility of our method.

## 2. Related Work

**Active Feature Acquisition (AFA).** Time-invariant AFA methods fall into two main classes:

*Greedy AFA policies:* These methods iteratively acquire features by greedily maximizing the expected information gain (Ma et al., 2018; Covert et al., 2023; Chattopadhyay et al., 2023). For example, Ma et al. (2018), Gong et al. (2019) and Chattopadhyay et al. (2022) use generative models to impute potential outcomes of all possible acquisitions and select

the greedy action. Covert et al. (2023) and Ghosh & Lan (2023) learn a policy network to directly predict the greedy action, guided by the loss of a separate prediction model. Gadgil et al. (2024) learns a value network to estimate the information gain directly. Theoretical work has shown that greedy policies achieve near-optimal performance compared to non-myopic ones under certain conditions (Golovin & Krause, 2011; Chen et al., 2015).

*Planning-based policies.* An alternative view treats AFA as a sequential decision-making problem addressed using reinforcement learning (RL). Model-based approaches learn a generative transition model using synthetic rollouts for data-efficient policy learning (Zannone et al., 2019; Li & Oliva, 2021). Model-free approaches directly learn value or Q-functions from offline data, selecting features that maximize expected returns (Shim et al., 2018; Kachuee et al., 2019; Janisch et al., 2019). However, these approaches are prone to model misspecification, given the challenges of offline value approximation and credit assignment over long acquisition trajectories (Erion et al., 2022).

**AFA under retrospective missingness.** Few studies examine how retrospective missingness affects feature acquisitions (Ma & Zhang, 2021; von Kleist et al., 2023). Most prior work either assumes fully observed data or uses simple imputation strategies such as conditional mean imputation, inducing statistical bias in policy evaluation (von Kleist et al., 2023). Model-based approaches provide a principled alternative when missingness assumptions hold, but face practical limitations: task-specific generative models are hard to estimate with limited data, particularly in high dimensions (Zannone et al., 2019; Li & Oliva, 2021; 2024).

**Meta-learning via sequence modeling.** Our proposed solution formulates feature acquisition as an in-context decision-making problem using sequence modeling, thereby avoiding explicit generative modeling assumptions. A growing body of work connects sequence models for meta-learning and in-context learning (ICL), to Bayesian inference (Müller et al., 2021; Nguyen & Grover, 2022; Ye & Namkoong, 2024), and extends this perspective to decision-making problems (Lee et al., 2023; Lin et al., 2024; Tianhui Cai et al., 2024; Moeini et al., 2025). In contrast, the AFA problem requires learning acquisition policies from offline data without directly observing reward-maximizing actions.

# 3. The Meta-Active Feature Acquisition Problem

We formalize the meta-AFA problem, where a single agent is trained across a distribution of tasks and generalizes to unseen tasks at inference time without per-task retraining. Let $\mathcal{T}$ denote a family of supervised learning tasks. We assume an unknown probability measure $\mathbb{P}$ over $\mathcal{T}$, from which indi-

vidual tasks $\tau \sim \mathbb{P}$ are sampled. For each task $\tau$, data pairs $(X, Y) \sim P_\tau(X, Y)$ are sampled from a task-specific distribution, where $X \in \mathbb{R}^{d_\tau}$ is a $d_\tau$-dimensional feature vector and $Y \in \mathcal{Y}$ is the target variable. We assume the features are time-invariant, i.e., the values of $X$ do not evolve over time. Let $X_j$ denote the value of feature $j$ and let $X_0$ denote the baseline features that are always observed. At each acquisition step $t \in \{1, \ldots, T\}$, the agent selects an action $A_t \in \{1, \ldots, d_\tau\}$, corresponding to the index of the next feature to acquire. We denote by $\underline{X}_t = \{X_0, \ldots, X_{A_t}\}$ the set of acquired features up to and including step $t$. For notational simplicity, we omit the task subscript $\tau$ and write $d$ when unambiguous, noting that the feature and action space are task-dependent. Throughout, random variables are written in uppercase and their realizations in lowercase.

Given a new task, *meta-active feature acquisition (meta-AFA)* selects at each step $t$ the next feature to acquire based on the partially observed features $\underline{X}_t$, with the objective of reducing predictive uncertainty on the target variable $Y$. We consider a fixed budget $b < d$ and for simplicity, uniform feature costs.

One common approach to (task-specific) AFA is to acquire features *greedily* based on the *expected* reduction in uncertainty, an approach rooted in Bayesian experimental design (Bernardo, 1979). At each step $t$, the method acquires the feature $X_j$ that maximizes the conditional mutual information (CMI) with the target:

$$
\begin{aligned}
I_\tau(Y; X_j \mid \underline{X}_t = \underline{x}_t) &\triangleq \\
\mathbb{E}_{X_j \mid \underline{x}_t} \left[ D_{\mathrm{KL}}\big( P_\tau(Y \mid X_j \cup \underline{x}_t) \,\|\, P_\tau(Y \mid \underline{x}_t) \big) \right].
\end{aligned}
\tag{1}
$$

Computing this quantity requires modeling the one-step conditional distributions $P_\tau(X_j|\underline{X}_t)$, $P_\tau(Y|\underline{X}_t, X_j)$ and $P_\tau(Y|\underline{X}_t)$. Reliable estimation of these conditionals is challenging because retrospective missingness leads to non-uniform coverage of feature combinations in the training set, a challenge we formalize in the following section.

## 3.1. Tasks with Retrospective Missingness

In practice, datasets often exhibit missingness with task-dependent mechanisms and rates, which can significantly affect the performance of AFA methods. To address this, we extend the meta-AFA problem to settings with retrospective missingness. In this setting, CMI (Equation (1)) is only identifiable and estimable under specific conditions. We formalize these conditions using a causal identifiability framework based on potential outcomes (Rubin, 1976), establishing when the distributions required for CMI estimation can be recovered from retrospectively missing data.

Let $R \in \{0, 1\}^d$ denote the binary indicators of feature observability (Nabi et al., 2020). $X(1)$ is the "potential outcome" of $X$, had $R = 1$ been true, i.e., the value that

would have been observed had the feature been measured. For a given $X_j(1) \in X(1)$ and $R_j \in R$, the realized feature value is generated by the following deterministic feature revelation mechanism:

$$X_j = \begin{cases} X_j(1) & \text{if } R_j = 1 \\ \text{``?''} & \text{if } R_j = 0 \end{cases}$$

Accordingly, the CMI estimand (Equation 1) can be equivalently written as:

$$I_\tau(Y; X_j \mid \underline{X}_t = \underline{x}_t) \equiv I_\tau(Y; X_j(1) | \underline{X}_t = \underline{x}_t), \quad (2)$$

and must be estimated under the joint distribution in the absence of missingness, $P_\tau(X(1), Y)$, which we refer to as the *reference distribution*. Identifying the reference distribution, $P_\tau(X(1), Y)$, from retrospectively missing data requires assumptions on the missingness mechanism, which are closely analogous to standard conditions used in off-policy evaluation. We formalize them as follows:

**Assumption 3.1.** (*Missing at Random or MAR*) $R_j \perp\!\!\!\perp X_j(1) \mid \underline{X}_t$.

**Assumption 3.2.** (*No Direct Effect*) $R_j \perp\!\!\!\perp Y \mid X_j(1), \underline{X}_t$.

**Assumption 3.3.** (*Positivity*) $P_\tau(R_j = 1 \mid \underline{X}_t = \underline{x}_t) > 0$ for all values $\underline{x}_t$ and $j \in \{1, ..., d\}$

Intuitively, Assumption 3.1 posits that any systematic differences between observed and missing data can be fully explained by the observed features, rather than by unobserved confounders. Assumption 3.2 states that measuring a feature does not directly affect the target variable. Assumption 3.3 requires sufficient data coverage of each feature acquisition action. Together, these assumptions yield the following identification result.

**Theorem 3.4.** (*Identification of CMI with retrospective missingness*) *The CMI for any subset $\underline{X}_t \subseteq X$ given by $I_\tau(Y; X_j(1) | \underline{X}_t = \underline{x}_t)$ is identified when $P_\tau(X(1), Y)$ is identified. Under Assumption 3.1 (MAR), 3.2 (No Direct Effect), and 3.3 (Positivity), the CMI can be estimated by*

$$I_\tau(Y; X_j(1) | \underline{X}_t = \underline{x}_t) = I_\tau(Y; X_j | \underline{X}_t = \underline{x}_t, R_j = 1)$$

The proof is provided in Appendix A.1. Intuitively, once the reference distribution, $P_\tau(X(1), Y)$, is identified, any functional of this distribution is also identified. However, estimating these functionals *directly from complete cases* ($R_j = 1$) at step $t$ is valid only under the stated assumptions. This restriction arises from requiring pointwise identification of the CMI-optimal acquisition action for every $\underline{x}_t$. In practice, such generality is often unnecessary, since many states are never encountered.

# 4. Method

We now present **L2M**, an end-to-end AFA framework that uses amortized optimization to meta-learn an acquisition policy with a transformer-based sequence model (Vaswani et al., 2017). We begin by introducing our proposed Bayesian analogue of the CMI objective in Theorem 3.4, using sequence models. Since the CMI objective is generally intractable, we formulate a tractable surrogate optimization problem that approximates the CMI objective. Finally, we relax the discrete action-selection problem with a smooth, differentiable approximation, enabling direct policy learning via gradient-based optimization.

## 4.1. Test-Time Feature Acquisition via Meta-Learned Sequence Models

At test time, we observe a context datase $\mathcal{D}_\tau^{\text{Context}} = \{(X_i, R_i, Y_i)\}_{i=1}^m$ consisting of $m \in \{1, .., N\}$ context samples drawn from an unknown task-specific distribution $P_\tau(X, R, Y)$, where $N$ is the maximum context length. Alongside this task-specific context, we are also given $M - m$ query samples, indexed by $q \in \{m + 1, \ldots, M\}$, from the same task distribution, whose targets $Y^{(q)}$ are unobserved and must be predicted. For each query instance $q$ at acquisition step $t$, the features are only partially observed, denoted by $\underline{X}_t^{(q)}$. During test-time, the objective is to sequentially acquire features for each query instance to infer its target, utilizing both its currently observed features $\underline{X}_t^{(q)}$ and the task-specific context provided by $\mathcal{D}_\tau^{\text{Context}}$.

The key challenge of meta-AFA is to model the joint predictive distribution over the unobserved query targets:

$$P(\text{Query Targets} \mid \text{Query States}, \text{Context Samples})$$
$$\triangleq P\left(Y^{m+1:M} \mid \underline{X}_t^{m+1:M}, \mathcal{D}_\tau^{\text{Context}}\right)$$

with sufficient flexibility while providing principled uncertainty estimates to guide feature acquisition. Sequence modeling offers a compelling solution: instead of explicitly modeling latent variables, autoregressive training over *data sequences*, together with invariance-inducing inductive biases (Definitions A.9, A.10), enables practical approximation of pointwise posterior predictive distributions (PPD) directly from observations. This perspective builds on prior works that formalize the connection between sequence models and Bayesian inference (Nguyen & Grover, 2022; Ye & Namkoong, 2024; Mittal et al., 2026). We note that the sequence model can be meta-learned using either synthetically generated tasks (Müller et al., 2021) or real-world data (Gardner et al., 2024).

We assume that all historical and query samples belonging to a given task are drawn independent and identically distributed (i.i.d.) from the underlying task distribution $P_\tau$. Consequently, the query instances are conditionally

independent given the task identity. Under this assumption, sequence modeling decomposes the joint predictive distribution over query samples into a product of one-step conditional probabilities,

$$P(Y^{m+1:M} \mid \underline{X}_t^{m+1:M}, \mathcal{D}_\tau^{\text{Context}})$$
$$= \prod_{q=m+1}^{M} P(Y^{(q)} \mid \underline{X}_t^{(q)}, \mathcal{D}_\tau^{\text{Context}}) \quad (3)$$

where we note that $P(Y^{(q)} \mid \underline{X}_t^{(q)}, \mathcal{D}_\tau^{\text{Context}})$ are pointwise PPDs. Intuitively, conditioning on a variable-length context of historical data allows the sequence model to infer the task-specific data-generating mechanism and amortize uncertainty estimation across partially observed queries. Once the joint predictive distribution is recovered via these one-step conditional probabilities, the naive strategy is to acquire the features with the maximum CMI given by

$$I(Y^{(q)}; X_j^{(q)} \mid \underline{X}_t^{(q)} = \underline{x}_t, R_j^{(q)} = 1, \mathcal{D}_\tau^{\text{Context}}) \quad (4)$$

However, directly maximizing this CMI is impractical because it requires access to the true step-wise conditional distributions and expectations over all candidate feature $X_j^{(q)}$. In the following section, we describe our methodology for constructing a tractable surrogate optimization problem using learned approximations.

### 4.2. Meta-Learning the Predictor and Acquisition Policy

Rather than computing the CMI exactly, we adopt the practical approximation of Covert et al. (2023, Prop. 2): we optimize the *one-step-ahead* predictive loss of a predictor $f_\phi$ for $Y$ after acquiring a candidate feature $X_j$. However, instead of a standard predictor, $f_\phi$ is a meta-predictor that outputs an approximation of the PPDs in Equation 3. A meta-policy $\pi_\theta$ is then trained to directly minimize this one-step loss, providing a tractable surrogate for the CMI objective. We now describe how both the predictor and the acquisition policy are meta-learned from training data.

To enable gradient-based optimization, we model the acquisition policy as a stochastic categorical distribution $\pi_\theta(\cdot \mid \underline{X}_t^{(q)}, \mathcal{D}_\tau) \in \Delta^{d-1}$, where $\Delta^{d-1}$ denotes the probability simplex over the feature acquisition action space and $\mathcal{D}\tau$ denotes the pretraining dataset. During training, we further restrict the feature acquisition policy to "blocked policies", which prevents the acquisition of features that are unavailable in the retrospectively observed training data.

**Definition 4.1** (Blocked Policy). A blocked policy $\tilde{\pi}_\theta$ is a stochastic feature acquisition policy that, at each step $t$, assigns non-zero probability only to features that have support in retrospectively observed data, i.e.,

$$\tilde{\pi}_\theta(j \mid \cdot) = 0 \quad \text{if} \quad R_j = 0,$$

where $R_j = 1$ indicates that feature $j$ is available.

**Greedy policy optimization** Restricting attention to blocked policies, we define a surrogate objective for jointly training the policy and predictor. Denote the state for the $q$-th sample in the sequence as $S_t^{(q)} = (\underline{X}_t^{(q)}, \underline{A}_{t-1}^{(q)})$, and $A_t$ is an action sampled from $\tilde{\pi}_\theta(S_t^{(q)}, \mathcal{D}_\tau^{1:m})$, where $\mathcal{D}_\tau^{1:m}$ denotes the first $m$ samples of pretraining data used as context. Let the per-action expected loss for a given query state $s_t^{(q)}$ be defined as:

$$J(a_t; s_t^{(q)}, \mathcal{D}_\tau^{1:m}) =$$
$$\mathbb{E}_{\substack{X_{a_t}^{(q)}, Y^{(q)} \mid \underline{s}_t^{(q)}, \\ \mathcal{D}_\tau^{1:m}, R_{a_t} = 1}} \left[ \ell\big(f_\phi(\cdot \mid s_t^{(q)} \cup X_{a_t}^{(q)}, \mathcal{D}_\tau^{1:m}), Y^{(q)}\big) \right]$$

where we use $\ell$ to denote the loss (e.g., negative log-likelihood) for evaluating the predictor. This yields the overall sequence prediction objective. To compute the training objective, we sample a task, $\tau$, draw a fixed $N$-length dataset $\mathcal{D}_\tau^N$ from that task. We then compute the cumulative sequence loss by iterating over context sizes $m \in \{1, \ldots, N-1\}$, treating the first $m$ samples as context $\mathcal{D}_\tau^{1:m}$ and the $(m+1)$-th sample as the query.

$$\mathcal{L}(f_\phi, \tilde{\pi}_\theta) =$$
$$\mathbb{E}_{\substack{\tau \sim \mathcal{T} \\ \mathcal{D}_\tau^N \sim P_\tau}} \left[ \sum_{m=1}^{N-1} \mathbb{E}_{S_t^{(q)}, R} \left[ \mathbb{E}_{A_t \sim \tilde{\pi}} \big[ J(A_t; s_t^{(q)}, \mathcal{D}_\tau^{1:m}) \big] \right] \right] \quad (5)$$

The query states $S_t^{(q)}$ and corresponding feature availability $R$ are jointly sampled from $P_\tau$. In practice, we sample $S_t^{(q)}$ uniformly from the set of available transitions within the $(m+1)$-th sample in $\mathcal{D}_\tau^N$. Our main theorem shows that minimizing the above sequence loss recovers the CMI maximizing actions.

**Theorem 4.2.** *(Sketch) Consider the sequence objective in Eq. 5 with cross-entropy loss and task-specific context $\mathcal{D}_\tau^{1:m} = (X^{1:m}, R^{1:m}, Y^{1:m})$. Any joint minimizer $(\theta^\star, \phi^\star)$ satisfies:*

1. **Bayes-optimal prediction:** $f_{\phi^\star}(\cdot \mid \underline{X}_t^{(q)}, \mathcal{D}_\tau^{1:m}) = p(Y^{(q)} \mid \underline{X}_t^{(q)}, \mathcal{D}_\tau^{1:m})$ *for all* $q$.
2. **CMI optimal acquisition:** $\pi_{\theta^\star}(\cdot \mid \underline{x}_t^{(q)}, \mathcal{D}_\tau^{1:m})$ *assigns probability only to*

$$\arg \max_{a:\, R_a^{(q)} = 1} I\Big(Y^{(q)}; X_a^{(q)} \mid \underline{x}_t^{(q)}, \mathcal{D}_\tau^{1:m}\Big),$$

*and is a point mass when the maximizer is unique.*

*Full theorem and proof are in Appendix A.3.*

Direct optimization of Equation 5 is non-differentiable due to the discrete nature of the action $A_t$, which is sampled from a categorical distribution. To obtain gradients with respect to the policy parameters, we use the straight-through Gumbel–Softmax relaxation of the sampling operation

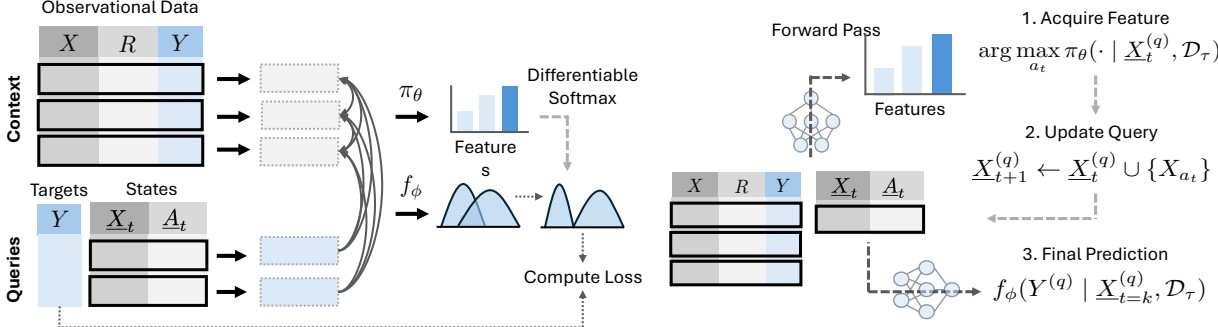

*Figure 2. Left:* Pretraining procedure for the predictor $f_\phi$ and policy $\pi_\theta$. From the pretraining task prior, we sample the data-generating mechanisms for the features, labels, and missingness, and sample a dataset. A Transformer model takes the dataset and partially observed query states $(\underline{X}_t^{(q)}, \underline{A}_t^{(q)})$ as input and predicts the corresponding targets $Y^{(q)}$ and the optimal action $A^{(q)}$. We train end-to-end with an autoregressive sequence loss over queries, where the loss is backpropagated through the policy's differentiable sampling operation. *Right:* The trained Transformer model is able to predict optimal acquisition actions for unseen query tasks in-context. We perform $k$ forward passes of the model, updating the state each time after a feature is acquired online. The attention mask to enable efficient autoregressive training under permutation invariance is given in Appendix A.6.

$A_t \sim \tilde{\pi}_\theta$. This relaxation replaces the non-differentiable categorical sample with a differentiable reparameterization, yielding a biased but low-variance gradient estimator that is more stable than zeroth-order methods such as REIN-FORCE (Williams, 1992; Maddison et al., 2016; Mohamed et al., 2020).

Specifically, at each step $t$, we draw Gumbel noise $\epsilon_t \sim$ Gumbel$(0, 1)$ and form a relaxed action vector $\hat{a}_t = $ softmax$\left(\frac{\log \tilde{\pi}_\theta + \epsilon_t}{\eta}\right)$, where $\eta$ is the temperature parameter. In the forward pass, we obtain a discrete action $\tilde{A}_t$ by taking the hard (one-hot) sample of $\hat{a}_t$. We denote this combined operation by $\tilde{A}_t = g_\theta(\epsilon_t; S_t^{(q)}, \mathcal{D}^{1:m})$. In the backward pass, gradients are propagated through the continuous relaxation $\hat{a}_t$. Using this reparameterization, the gradient of Equation 5 admits the following pathwise approximation:

$$\nabla_\theta \mathcal{L}(f_\phi, \tilde{\pi}_\theta) =$$
$$\mathbb{E}_{\substack{\tau \sim \mathcal{T} \\ \mathcal{D}_\tau^N \sim P_\tau}} \left[ \sum_{m=1}^{N-1} \mathbb{E}_{S_t^{(q)}} \left[ \mathbb{E}_{\epsilon_t} \left[ \nabla_\theta J(\tilde{A}_t; s_t^{(q)}, \mathcal{D}_\tau^{1:m}) \right] \right] \right], \quad (6)$$

where the inner expectation is taken over the Gumbel noise. In practice, we approximate this expectation using Monte Carlo sampling by drawing $N_{\text{MC}}$ independent noise samples $\{\epsilon_t^{(i)}\}_{i=1}^{N_{\text{MC}}}$ and form the corresponding reparameterized action trajectories $\tilde{A}_t^{(i)} = g_\theta(\epsilon_t^{(i)}; S_t^{(q)}, \mathcal{D}_\tau^{1:m})$. The resulting pathwise gradient estimator is

$$\widehat{\nabla_\theta J} = \frac{1}{N_{\text{MC}}} \sum_{i=1}^{N_{\text{MC}}} \nabla_\theta J(\tilde{A}_t^{(i)}; s_t^{(q)}, \mathcal{D}_\tau^{1:m}) \quad (7)$$

**Planning-based policy optimization** In Appendix A.5, we extend the surrogate objective and the optimization proce-

dure to $k$-step lookahead policies that optimize for expected loss after $k + 1$ steps. In our experiments, we train one-step lookahead policies ($k = 1$) and $N_{MC} = 4$.

### 4.3. Model Architecture and Training

**Model Architecture.** While our conceptual framework and policy optimization are agnostic to the specific parameterization of sequence models, we propose a joint Transformer encoder with a separate target predictor and acquisition policy head as a proof of concept. Relative to a standard Transformer, we make the following key changes to model the posterior predictive distributions in Equation 5. (i) To represent missing features, each sample $i$ in a sequence is represented as the concatenation of its masked features, missingness indicator, and label: $[\mathbf{x}^i \odot \mathbf{r}^i, \mathbf{r}^i, y^i]$. The encoding strategy is standard in prior AFA work (Covert et al., 2023; Gadgil et al., 2024; Norcliffe et al., 2025). (ii) To make the model invariant to the ordering of samples in the sequence, we remove positional embeddings and replace causal masking with an alternative attention masking structure during both training and inference for permutation invariance, consistent with related work (Nguyen & Grover, 2022; Ye & Namkoong, 2024). Additional details of our architecture are given in Appendix A.6.

**Training.** In practice, our pretraining consists of two stages. In the first stage, we pretrain $f_\phi$ on query points with random feature subsets $X_o$ for any $o \subset [d]$, exploiting the fact that the optimal predictor is independent of the policy $\pi_\theta$.. In the second stage, we train $\pi_\theta$ with $f_\phi$ held fixed. At each training step, we sample a batch of tasks with different label distributions and missingness mechanisms from $\mathbb{P}_\mathcal{T}$, draw data for each task, and present the samples as a

randomly permuted sequence to the model. We implement blocked policies (Theorem 4.1) by masking unavailable actions to ensure that missing features (with $R_j = 0$) are not acquired by the policy during training. Details of the training procedure are provided in Algorithm 1 and Algorithm 2 (Appendix A.6). Figure 2 (left) summarizes the procedure for a single task.

## 5. Experiments

We aim to demonstrate the feasibility of our **L2M** framework across multiple tasks and diverse applications. While we provide comprehensive experimental details in the Appendix A.7, we provide a brief overview in this section.

**Simulated tasks.** First, we evaluate our framework on fully synthetic classification and regression tasks to validate uncertainty quantification and planning behavior across diverse prior specifications and varying feature dimensions. Following the pretraining procedure in Figure 2, we specify the full data-generating process for each synthetic task by sampling its feature distribution, conditional label model, and missingness mechanism.

Classification tasks are sampled from Bayesian neural network priors (Müller et al., 2021). To evaluate the benefit of lookahead loss, we additionally consider tasks with sparse pairwise feature synergies (**BNN-Plan**). Specifically, for selected pairs $(i, j)$, an interaction signal proportional to $x_i x_j$ becomes informative when $x_i x_j > \tau$. This thresholding induces interactions that can violate submodularity (diminishing returns), making greedy acquisition suboptimal.

For the regression setting, training tasks are sampled from a Gaussian Process (**GP**) prior with a zero-mean $m(x) = 0$ and randomized RBF kernels, denoted as $\tau_i \sim \mathcal{GP}(m, k)$. To access generalization, we sample evaluation GP tasks from both familiar RBF kernels, and unseen Matern kernels.

**Real-world tasks.** Next, we evaluate **L2M** on realistic tasks derived from real-world tabular datasets: **Metabric** (Curtis et al., 2012), **MiniBooNE** (Roe et al., 2005), and **MIMIC-IV** (Johnson et al., 2023). During pre-training, we construct semi-synthetic classification tasks by sampling labels from **BNN** without additional interactions. The feature distributions are obtained by sampling real instances and introducing a known synthetic missingness mechanism. We then evaluate model performance on tasks with varying sample sizes and degrees of synthetic missingness, using labels from the original datasets. This setup preserves realistic feature distributions while enabling a controlled assessment of model generalization to real-world tasks.

To further demonstrate the generality of our framework, we evaluate **L2M** on **MNIST**, where training task labels are drawn from real binary digit-pair tasks. Images are divided

into $d = 20$ candidate pixel blocks for acquisition. At each training step, we sample a binary classification task between two randomly chosen digits, training the model to adaptively differentiate between images of two digits in-context. We evaluate the performance on datasets with varying missingness and sample sizes, using unseen queries.

**Baselines.** Because prior AFA work is task-specific, we compare against AFA methods that train a separate model for each task. We focus on two greedy, CMI-based approaches: gradient dynamic feature selection (**GDFS** (Covert et al., 2023)), which uses MLPs instead of sequence modeling, and discriminative mutual information estimation (**DIME** (Gadgil et al., 2024)), which trains an MLP as a value network to estimate CMI directly. To ensure fair evaluation, both **L2M** and task-specific methods are evaluated on the same held-out tasks and query sets. We also include an RL baseline using Deep Q-learning (**DQN**) (Shim et al., 2018; Kachuee et al., 2019; Janisch et al., 2019), but exclude it from synthetic tasks due to the prohibitive computational cost of learning hundreds of test tasks. To ensure comparability with greedy strategies, we define the per-step reward as the log likelihood under the current predictor, and each trajectory terminates when all features are acquired. Additionally, we include meta-learning ablations in Appendix 18 where we evaluate conditional neural processes that learns a fixed task representation, and additional gradient-based task-specific fine-tuning.

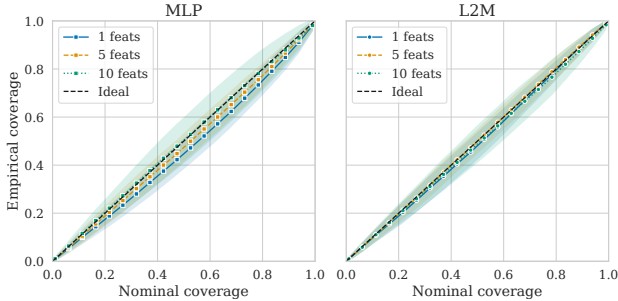

*Figure 3.* Coverage plots comparing MLP and **L2M** at various acquisition steps. **L2M** shows robust coverage compared to the MLP. The dashed diagonal line denotes perfect calibration. Across the acquisition trajectory, **L2M** remains closer to the ideal line than the MLP baseline. Shaded regions indicate standard error across 200 sampled evaluation tasks. To ensure a fair comparison, both methods are evaluated on the same random acquisition trajectories using consistent evaluation tasks and samples.

### 5.1. Results

**L2M achieves more reliable uncertainty quantification of target variable prediction than task-specific models.** We show that meta-learning across diverse tasks enhances uncertainty quantification relative to task-specific baselines. We evaluate uncertainty quality using three complementary

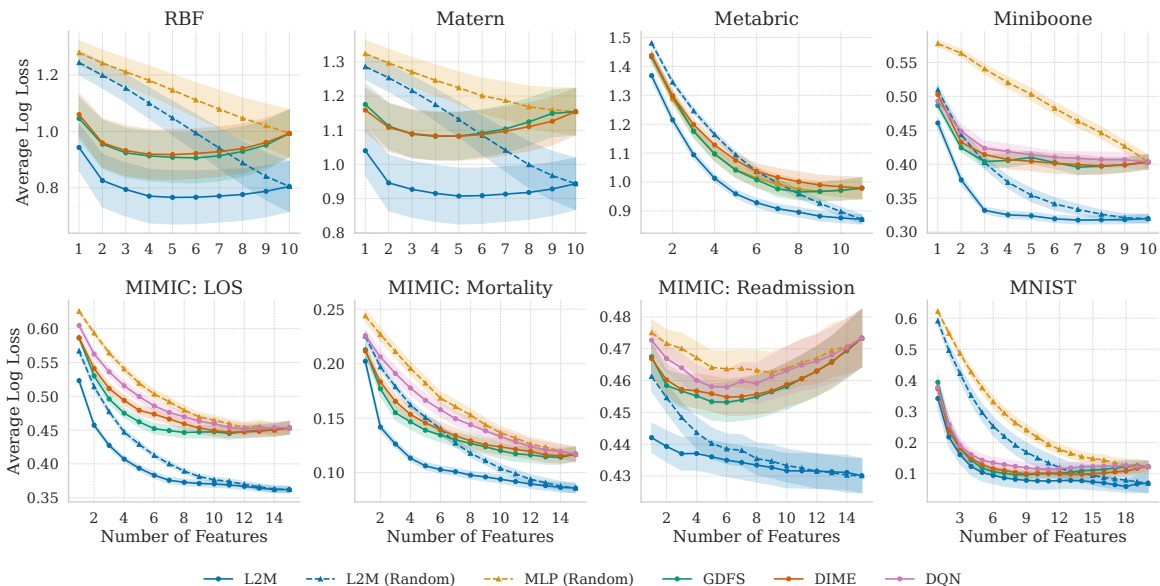

*Figure 4.* Acquisition performance measured by log loss (lower is better), averaged over tasks derived from various synthetic and real-world datasets. RBF and Matérn results demonstrate that our meta-learning approach generalizes to both in-distribution and out-of-distribution tasks, respectively. MIMIC-IV results show that a single pretrained L2M model generalizes to diverse tasks with real, unseen labels. In many cases, acquisition performance also surpasses task-specific greedy and RL approaches. Additional metrics are provided in Appendix Figure 10.

metrics: empirical vs. nominal coverage (Kompa et al., 2021), log loss, and mean squared error (MSE) (Nguyen & Grover, 2022). Figure 3 shows that **L2M** outperforms task-specific MLP baselines in average task-level coverage for synthetic GP regression tasks across an acquisition trajectory of 1 to 10 features. Additional analysis provided in Appendix Figure 9 shows that **L2M** also achieves lower log loss and MSE. Additional results on semi-synthetic and real-world classification tasks are reported in Appendix Figure 7 and 8. We attribute this improvement in uncertainty quantification to **L2M** leveraging its diverse learned task prior to mitigate the effects of reduced coverage. This improvement in uncertainty quantification is critical, as it directly translates to stronger downstream acquisition performance for both the RBF and Matérn kernel tasks, as shown in Figure 4.

**L2M pretraining induces adaptive acquisition behavior across evaluation tasks.** Next, we demonstrate empirical evidence that our proposed pretraining approach allows **L2M** to learn feature acquisition policies that generalize to a diverse set of tasks. We use log loss at each feature acquisition step to evaluate whether the policy is successfully reducing uncertainty about the target variable. Figure 4 demonstrates that (1) **L2M** achieves lower log loss compared to random acquisition from the same model and (2) **L2M** consistently achieves lower log loss than all task-specific baselines on evaluation tasks across datasets (additional metrics are shown in Appendix Figure 10). The magnitude of these gains varies by dataset, depending on how well the syn-

thetic or semi-synthetic pretraining task prior aligns with the downstream task, and the inherent task complexity of the real-world tasks. Nonetheless, **L2M** 's advantage over task-specific AFA methods is consistent, highlighting the value of reliable uncertainty quantification learned through meta-training on diverse tasks. In contrast, task-specific AFA baselines degrade as more features are acquired, due to the difficulties of jointly learning predictors and acquisition policies from limited data. A key finding is that **L2M** delivers the *largest benefits in regimes with shorter contexts and higher rates of retrospective missingness* compared to task-specific models, as shown in Appendix Figure 11. Autoregressive meta-training enables reliable uncertainty propagation across different context lengths, yielding robustness under varying acquisition budgets. Furthermore, we hypothesize that robust performance across varying rates of retrospective missingness arises from our sequence modeling framework, which explicitly propagates the uncertainty induced by missing historical data rather than collapsing it through deterministic point imputations. These properties are especially useful in healthcare, where labeled data are scarce and missingness is prevalent.

**L2M can be trained using lookahead signals for multi-step planning.** We demonstrate that our lookahead variant meta-learns policies that can match or potentially improve over greedy acquisition under a diverse fully-synthetic task priors (Figure 5). We further show that our pathwise estimation framework leads to improved sample efficiency and

lower error in gradient estimation for greedy and lookahead objectives when meta-training on diverse tasks (Appendix Figures 17).

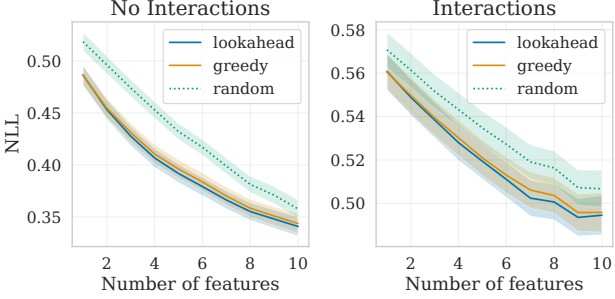

*(a)* NLL across acquisition steps

*Figure 5.* We compare **L2M** trained on synthetic tasks with a greedy objective to a one-step lookahead variant. (a) Evaluation is on 200 synthetic tasks where (i) greedy acquisition is sufficient and (ii) pairwise feature synergies make lookahead beneficial. The lookahead variant outperforms greedy in both settings, with the largest gains emerging at later acquisition steps.

**L2M pretrained on fully synthetic tasks generalizes to unseen real-world tasks.** We provide additional results demonstrating that our **L2M** framework can be trained on a fully synthetic task prior with synthetically generated covariate distributions, and still generalize beyond the training distribution. We show that under this challenging setting, the model (1) learns to acquire features effectively on in-distribution synthetic tasks, demonstrating comparable performance to task-specific models (Appendix Figure 13) and (2) learns policies that outperform random acquisition when performing in-context learning on unseen real-world data distributions (Appendix Figure 14). We attribute this generalization to the ability of in-context learning to identify the closest mechanism from its pretraining prior when faced with unseen tasks (Nagler, 2023). We futher show that task-specific finetuning improves performance and mitigates prior mismatch for the fully synthetic variant (Appendix Figure 14). Finally, we provide stress tests using unseen real-world missingness patterns in **MIMIC-IV**, showing that synthetic pretraining can still identify policies that improve over random acquisition on out-of-distribution real-world tasks (Appendix Figure 16).

## 6. Discussion

**Conclusion.** In this work, we introduced the meta-AFA problem and presented **L2M**, an end-to-end differentiable, uncertainty-driven framework for feature acquisition that performs in-context learning across tasks. Our contributions are threefold. First, we formalized the meta-AFA problem and established the theoretical assumptions under which meta-learning acquisition policies is feasible. Second, we developed an algorithm that enables in-context active fea-

ture acquisition, leveraging meta-learned approximations of target conditional distributions for AFA policy optimization. Third, we empirically validated our framework across synthetic and real-world settings, demonstrating reliable uncertainty estimation and principled acquisition behavior under both in-distribution and out-of-distribution conditions compared to task-specific baselines. Notably, **L2M** exhibits robust performance under limited labeled data and significant retrospective missingness, conditions that pose substantial challenges for existing approaches. We believe that the meta-AFA formulation opens promising avenues for future research, including extending the framework to broader data modalities, and scaling pretraining to richer task distributions and missingness patterns.

**Limitations** *(i)* Our approach relies on sufficient offline action coverage and MAR, an untestable but realistic assumption about the missingness mechanism. Future work will relax these assumptions and investigate whether our uncertainty estimates can serve as informative bounds or diagnostics for violations of positivity or MAR (Jesson et al., 2020). *(ii)* Empirically, we demonstrate the utility with tabular models pretrained from scratch on simple synthetic task priors. Scaling to diverse, large-scale real datasets across domains, while plausible, may require principled prior-specification procedures. Leveraging priors encoded in pretrained language models is another promising direction. *(iii)* We do not consider solutions for the full cost-sensitive MDP formulation, which requires multi-step planning for long-term reward. We discuss limitations of short-horizon (i.e., greedy) acquisition in Appendix A.4. *(iv)* While we demonstrate proof-of-concept on medium-scale tabular datasets, extension to large tabular datasets is plausible using larger pretraining architectures, such as using TabPFN (Grinsztajn et al., 2025; Hollmann et al., 2025; 2023) with additional compute resources. *(v)* Finally, we restrict attention to time-invariant settings; extending to time-varying dynamics is a crucial aspect of future work.

## Impact Statement

This paper introduces a meta-learning framework for sequentially acquiring information under uncertainty, allowing models to learn acquisition strategies that generalize across a distribution of tasks. The approach can improve the efficiency of prediction systems by reducing unnecessary measurements while maintaining performance, which is relevant for domains like clinical triage. Our work is a proof of concept, and we do not claim deployment readiness: real-world use would require prospective validation, careful specification of costs and utilities, and human oversight.

# Acknowledgements

YK acknowledges partial support from the Precision Psychiatry and Mental Health Pilot Award at Columbia. YK, ZJ, and SJ acknowledge partial support from 5R01MH137679-02. ZJ and SJ acknowledge partial support from the Google Research Scholar Award. JY acknowledges support from the Columbia University Data Science Institute. Any opinions, findings, conclusions, or recommendations in this manuscript are those of the authors and do not reflect the views, policies, endorsements, expressed or implied, of any aforementioned funding agencies/institutions.

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

# A. Appendix

## A.1. Proof of Theorem 3.4

**Theorem 3.4** The CMI objective under missing data for a candidate feature $j$ given the current set of observed features $\underline{x}_t$ is given by the following:

$$
\begin{aligned}
&I(Y; X_j(1) \mid \underline{X}_t = \underline{x}_t) \\
&= \sum_{y, x_j(1)} p(y, x_j(1) \mid \underline{x}_t) \log \frac{p(y, x_j(1) \mid \underline{x}_t)}{p(y \mid \underline{x}_t) p(x_j(1) \mid \underline{x}_t)} \\
&= \sum_{y, x_j(1)} p(y \mid x_j(1), \underline{x}_t) p(x_j(1) \mid \underline{x}_t) \log \frac{p(y \mid x_j(1), \underline{x}_t) p(x_j(1) \mid \underline{x}_t)}{p(y \mid \underline{x}_t) p(x_j(1) \mid \underline{x}_t)} \\
&= \sum_{y, x_j(1)} p(y \mid x_j(1), \underline{x}_t, R_j = 1) p(x_j(1) \mid \underline{x}_t, R_j = 1) \log \frac{p(y \mid x_j(1), \underline{x}_t, R_j = 1) p(x_j(1) \mid \underline{x}_t, R_j = 1)}{p(y \mid \underline{x}_t) p(x_j(1) \mid \underline{x}_t, R_j = 1)} \\
&= \sum_{y, x_j} p(y \mid x_j, \underline{x}_t, R_j = 1) p(x_j \mid \underline{x}_t, R_j = 1) \log \frac{p(y \mid x_j, \underline{x}_t, R_j = 1) p(x_j \mid \underline{x}_t, R_j = 1)}{p(y \mid \underline{x}_t) p(x_j \mid \underline{x}_t, R_j = 1)} \\
&= \sum_{y, x_j} p(y, x_j \mid \underline{x}_t, R_j = 1) \log \frac{p(y, x_j \mid \underline{x}_t, R_j = 1)}{p(y \mid \underline{x}_t) p(x_j \mid \underline{x}_t, R_j = 1)} \\
&= I(Y; X_j \mid \underline{X}_t = \underline{x}_t, R_j = 1),
\end{aligned}
$$

The third equality holds due to the MAR assumption $R_j \perp\!\!\!\perp X_j(1) \mid \underline{X}_t$ and no direct measurement effect $R_j \perp\!\!\!\perp Y \mid X_j(1), \underline{X}_t$. Positivity ensures all conditional distributions are well-defined.

## A.2. Proof of Proposition A.2

We begin by showing that minimizing the surrogate one-step loss given in Equation 8 for a single task recovers the greedy CMI actions. Our result is a slight modification from the result shown in (Covert et al., 2023).

The surrogate loss for a single task $\tau \sim \mathbb{P}_\mathcal{T}$ is given by:

$$
\mathcal{L}(\theta, \phi) = \mathbb{E}_{S_t, R \sim P_\tau} \left[ \mathbb{E}_{A_t \sim \tilde{\pi}(\cdot \mid S_t)} \left[ \mathbb{E}_{X_{A_t}, Y \mid \underline{X}_t, R_{A_t} = 1} \left[ \ell(f_\phi(\cdot \mid \underline{X}_t \cup X_{A_t}), Y) \right] \right] \right] \tag{8}
$$

where states are denoted as $S_t = (\underline{X}_t, \underline{A}_t)$ and the acquisition action is sampled from the blocked policy $A_t \sim \tilde{\pi}_\theta(\cdot \mid S_t)$. The availability of actions for the blocked policy is dictated by the missingness indicator $R$, where $(S_t, R)$ are jointly drawn from the task distribution $P_\tau$.

*Remark* A.1. For this theorem, the loss can average over the state distribution at step $t$, denoted $d_t^{\tilde{\pi}_\theta}(\cdot \mid \tau)$, obtained by either sampling uniformly over available states, or rolling out the (blocked) policy $\tilde{\pi}_\theta$ for $t$ steps from the initial law $d_0(\cdot \mid \tau)$ induced by $P_\tau(X, R, Y)$. The choice of $d_t$ is flexible (with caveats) and should have sufficient overlap with the intended deployment state distribution. Note that the per-state argmax (the optimal acquisition action at each step) does not change regardless of the outer state distribution.

**Proposition A.2.** *(Surrogate optimality for greedy CMI) For a given task $\tau$, consider the population objective (Eq. 8 with cross-entropy loss). Then any joint minimizer $(\theta^\star, \phi^\star)$ of $\mathcal{L}$ satisfies:*

1. *$\phi^\star$ is Bayes-optimal: $f_{\phi^\star}(\cdot \mid \underline{X}_t) = P_\tau(Y \mid \underline{X}_t)$;*

2. *$\pi_{\theta^\star}$ places all its mass on actions that maximize the conditional mutual information*

$$
j \in \arg\max_{j: R_j = 1} I_\tau\big(Y; X_j \mid \underline{X}_t = \underline{x}_t, R_j = 1\big),
$$

*i.e. $\pi_{\theta^\star}(j \mid \underline{x}_t) > 0$ only if $j$ is a greedy-CMI maximizer. If the maximizer is unique, $\pi_{\theta^\star}$ is a point mass on that action.*

*Proof.* Part I - Proof of Bayes-optimality:

We fix $\theta$ and consider the predictor. We begin with a standard fact: under cross-entropy loss for a discrete binary target, the conditional risk is minimized by the true conditional. In other words, to minimize expected loss the model $f_\phi$ needs to closely approximate the true distribution $P$. We show that this holds agnostic to the choice of $\tau$.

**Lemma A.3** (Bayes optimality under cross-entropy). *Let $\ell(q, y) = -\log q(y)$. Then the minimizer $\phi^\star$ satisfies*

$$f_{\phi^\star}(\underline{X}_t) = \arg\min_{f_\phi(\cdot)} \mathbb{E}_{Y|\underline{X}_t}[\ell(f_\phi(\cdot \mid \underline{X}_t), Y)] = P(Y \mid \underline{X}_t).$$

*Furthermore, this minimizer does not depend on $\theta$. In particular, any $f_{\phi^\star}$ that matches the true conditional for all such $(\underline{x}_t, x_j)$ is a global minimizer for every policy.*

*Proof.* We denote $p(i \mid \underline{X}_t) = p_i$ as the conditional class probabilities and $f_i = f_\phi(i \mid \underline{X}_t)$ as the learner's predicted class probabilities, where $i \in \{0, 1\}$ are the binary class labels. The conditional risk decomposes as

$$\mathbb{E}_{Y|\underline{X}_t}[\ell(f_\phi(\cdot \mid \underline{X}_t), Y)] = -\sum_{i=0}^{1} p_i \log f_i$$

$$= -\sum_{i=0}^{1} p_i \log p_i + \sum_{i=0}^{1} p_i \log \frac{p_i}{f_i} = \underbrace{H(Y \mid \underline{X}_t)}_{\text{Constant}} + \text{KL}(P(Y|\underline{X}_t) \,\|\, f_\phi(Y|\underline{X}_t)).$$

$\square$

Part II - Proof of maximizer equivalence

Once we have Bayes optimality with the learner at $\phi^\star$, we can rewrite the inner risk as the expected conditional entropy using the following lemma:

**Lemma A.4** (Risk reduces to (expected) conditional entropy at $\phi^\star$). *With $\ell(q, y) = -\log q(y)$ and $f_{\phi^\star}$ as in Lemma A.3, for any task $\tau$ and step $t$, any history $\underline{x}_t$ and retrospective feature availability $r_j$*

$$\mathbb{E}_{Y, X_j | \underline{x}_t, R_j = r_j}[\ell(f_{\phi^\star}(\cdot \mid \underline{x}_t \cup X_j), Y)] = \mathbb{E}_{X_j | \underline{x}_t, R_j = r_j}\big[H_\tau(Y \mid \underline{x}_t, X_j)\big].$$

*Consequently, the policy-evaluated inner term in equation 8 is*

$$\mathbb{E}_{A_t \sim \tilde{\pi}_\theta(\cdot|s_t)} \mathbb{E}_{X_{a_t} | \underline{x}_t, R_{a_t} = 1}\big[H_\tau(Y \mid \underline{x}_t, X_{a_t})\big].$$

*Proof.* Performing the similar decomposition as in Lemma A.3,

$$\mathbb{E}_{Y, X_j | \underline{x}_t, r_j}\big[\ell(f_{\phi^\star}(\cdot \mid \underline{x}_t \cup X_j), Y)\big]$$

$$= \mathbb{E}_{X_j | \underline{x}_t, r_j}\Big[\mathbb{E}_{Y | \underline{x}_t, X_j}\big[-\log f_{\phi^\star}(Y \mid \underline{x}_t, X_j)\big]\Big]$$

$$= \mathbb{E}_{X_j | \underline{x}_t, r_j}\Big[H_\tau(Y \mid \underline{x}_t, X_j) + \text{KL}\big(P_\tau(Y \mid \underline{x}_t, X_j) \,\|\, f_{\phi^\star}(Y \mid \underline{x}_t, X_j)\big)\Big]$$

$$= \mathbb{E}_{X_j | \underline{x}_t, r_j}\big[H_\tau(Y \mid \underline{x}_t, X_j)\big]$$

Where the first equality follows from iterated expectations. In the last equality, the KL term vanishes at $\phi^\star$. $\square$

Now, we consider the loss in equation 8. Plugging $\phi^\star$ and using Lemma A.4 for the given choice of task $\tau$,

$$\mathcal{L}(\theta, \phi^\star) = \mathbb{E}_{S_t, R \sim P_\tau}\left[\mathbb{E}_{A_t \sim \tilde{\pi}_\theta(\cdot|s_t)} \mathbb{E}_{X_{a_t} | \underline{x}_t, R_{a_t} = 1}\big[H_\tau(Y \mid \underline{x}_t, X_{a_t})\big]\right]$$

$$\overset{\text{blocked}}{=} \mathbb{E}_{S_t, R \sim P_\tau}\left[\sum_{\{a_t : R_{a_t} = 1\}} \tilde{\pi}_\theta(a_t \mid s_t) \mathbb{E}_{X_{a_t} | \underline{x}_t, R_{a_t} = 1}\big[H_\tau(Y \mid \underline{x}_t, X_{a_t})\big]\right]$$

where $\{a_t : R_{a_t} = 1\}$ is the set of available features at step $t$. For fixed $\underline{x}_t$,

$$\sum_{a_t \in \{a_t : R_{a_t} = 1\}} \pi(a_t | s_t) \, \mathbb{E}_{X_{a_t} | \underline{x}_t, R_{a_t} = 1}[H_\tau(Y \mid \underline{x}_t, X_{a_t})]$$

is linear over the simplex on $\{a_t : R_{a_t} = 1\}$ and is therefore minimized by placing all mass on

$$\arg \min_{a_t \in \{a_t : R_{a_t} = 1\}} \mathbb{E}_{X_{a_t} | \underline{x}_t, R_{a_t} = 1}[H_\tau(Y \mid \underline{x}_t, X_{a_t})] \,.$$

Since $H(Y \mid \underline{x}_t)$ does not depend on $a_t$, we can obtain the optimal action among the available actions

$$a_t^* = \arg \min_{a_t \in \{a_t : R_{a_t} = 1\})} \mathbb{E}_{X_{a_t} | \underline{x}_t, R_{a_t} = 1}[H_\tau(Y \mid \underline{x}_t, X_{a_t})] = \arg \max_{a_t \in \{a_t : R_{a_t} = 1\})} I_\tau(Y; X_{a_t} \mid \underline{x}_t, R_{a_t} = 1) \,,$$

because

$$I_\tau(Y; X_{a_t} \mid \underline{x}_t, R_{a_t} = 1) = H_\tau(Y \mid \underline{x}_t) - \mathbb{E}_{X_{a_t} | \underline{x}_t, R_{a_t} = 1}[H_\tau(Y \mid \underline{x}_t, X_{a_t})] \,.$$

We define the above $a_t^*$ as the locally optimal action. We can define the full set of all possible actions (features) as $\mathcal{A}_{\text{all}}$. Then, we define the globally optimal action $a_{\text{global}}^\star$, which is the action that maximizes the conditional mutual information if all actions were available:

$$a_{\text{global}}^\star = \arg \max_{a \in \mathcal{A}_{\text{all}}} I_\tau(Y; X_{a_t} \mid \underline{x}_t, R_{a_t} = 1)$$

Because the expectation is taken over the distribution $P_\tau(S_t, R)$, minimizing the expected loss requires minimizing the inner sum for every specific realization of $R$ that has non-zero probability given the state, $P_\tau(R \mid S_t) P_\tau(S_t)$. Let $\mathcal{R}$ be the set of all possible missingness indicator vectors. We partition $\mathcal{R}$ into two disjoint sets based on the availability of the global optimum, where we have $\mathcal{R}_{\text{opt}} = \{R \in \mathcal{R} : R_{a_{\text{global}}^\star} = 1\}$ is the set of missingness indicators where the globally optimal action is available, and $\mathcal{R}_{\text{sub}} = \{R \in \mathcal{R} : R_{a_{\text{global}}^\star} = 0\}$ is the set where the globally optimal action is missing.

Consider any realization of the missingness indicator $R \in \mathcal{R}_{\text{opt}}$. By definition, $R_{a_{\text{global}}^\star} = 1$, meaning $a_{\text{global}}^\star$ is in the set of available actions. Because the conditional entropy ranking (or $I_\tau$) of actions is globally consistent and invariant to the realization of $R$, the local minimizer over the available set is strictly the global minimizer $a_{\text{global}}^\star$. For realizations $R \in \mathcal{R}_{\text{sub}}$, the mass is similarly placed on the next available feature with the lowest conditional entropy. $\qquad \square$

### A.3. Proof of Theorem 4.2

Our main theorem extends Proposition A.2 to conditional distributions that incorporate task-specific context. We leverage the following conditional independence assumption, which improves tractability by removing the need to fully model the joint via an autoregressive factorization.

**Assumption A.5** (Conditional independence across queries)**.** Given the context $\mathcal{D}_\tau^{1:m}$ and the per-query partial inputs $\underline{X}_t^{m+1:N}$, the query targets are conditionally independent:

$$p\big(Y^{m+1:N} \mid \underline{X}_t^{m+1:N}, \mathcal{D}_\tau^{1:m}\big) = \prod_{q=m+1}^{N} p\big(Y^{(q)} \mid \underline{X}_t^{(q)}, \mathcal{D}_\tau^{1:m}\big) \,.$$

**Theorem 4.2** [Surrogate optimality for greedy CMI with context] Consider the sequence modeling objective in Eq. 9 with cross-entropy loss. Let $\mathcal{D}_\tau^{1:m} = (X^{1:m}, R^{1:m}, Y^{1:m})$ be the task-specific context and $\underline{X}_t^{m+1:N}$ the partially observed features for the $N - m$ query points at step $t$. Then any joint minimizer $(\theta^\star, \phi^\star)$ of $\mathcal{L}$ satisfies:

1. **Per-query Bayes-optimality.** For each $q \in \{m+1, \ldots, N\}$,

$$f_{\phi^\star}\big(\cdot \mid \underline{X}_t^{(q)}, \mathcal{D}_\tau^{1:m}\big) = p\big(Y^{(q)} \mid \underline{X}_t^{(q)}, \mathcal{D}_\tau^{1:m}\big) \,.$$

Consequently by assumption A.5 ,

$$f_{\phi^\star}\big(\cdot \mid \underline{X}_t^{m+1:N}, \mathcal{D}_\tau^{1:m}\big) = p\big(Y^{m+1:N} \mid \underline{X}_t^{m+1:N}, \mathcal{D}_\tau^{1:m}\big)$$

2. **Step-wise CMI-optimal acquisition for every query.** For each query index $q \in \{m+1, \ldots, N\}$ and for $(\underline{x}_t^{(q)}, \mathcal{D}_\tau^{1:m})$, the policy places mass only on actions that maximize CMI:

$$j \in \arg \max_{a_t : R_{a_t} = 1} I\left(Y^{(q)}; X_{a_t}^{(q)} \;\middle|\; \underline{X}_t^{(q)} = \underline{x}_t^{(q)}, \; R_{a_t}^{(q)} = 1, \; \mathcal{D}_\tau^{1:m}\right).$$

If the maximizer is unique, $\pi_{\theta^\star}(\cdot \mid \underline{x}_t^{(q)}, \mathcal{D}_\tau^{1:m})$ is a point mass on that action.

Part I - Proof of Bayes-optimality:

We first fix $\theta$ and consider the predictor. The loss is given by the following, where again the query states and corresponding action availability $(S_t, R)$ are jointly sampled from $P_\tau$:

$$\mathcal{L}(f_\phi, \pi_\theta) = \mathbb{E}_{\substack{\tau \sim \mathcal{T} \\ \mathcal{D}_\tau^N \sim P_\tau}} \left[ \sum_{m=1}^{N-1} \mathbb{E}_{S_t^{(q)}, R \sim P_\tau} \left[ \mathbb{E}_{A_t \sim \tilde{\pi}(\cdot | S_t^{(q)}, \mathcal{D}_\tau^{1:m})} [J(A_t; \underline{X}_t^{(q)}, \mathcal{D}_\tau^{1:m})] \right] \right]$$

where

$$J(a_t; \underline{x}_t^{(q)}, \mathcal{D}_\tau^{1:m}) = \mathbb{E}_{\substack{X_{a_t}^{(q)}, Y^{(q)} | \underline{x}_t^{(q)}, \\ \mathcal{D}_\tau^{1:m}, R_{a_t} = 1}} \left[ \ell\left( f_\phi(\cdot \mid \underline{x}_t^{(q)} \cup X_{a_t}^{(q)}, \mathcal{D}_\tau^{1:m}), Y^{(q)} \right) \right]$$

is the per-action expected loss for a given query feature set $x_t^{(q)}$ and length-$m$ context $\mathcal{D}_\tau^{1:m}$.

We show Bayes-optimality by showing that to minimize expected loss, $f_\phi$ needs to closely approximate the true distribution, analogous to Lemma A.3.

*Proof.* Without loss of generality, we fix the context length $m$. We consider the minimizer $\phi^\star$ of the loss summed over each query $q \in \{m+1, ..., N\}$. Lemma A.3 applied to each query shows that this loss recovers the per-query conditional i.e.

$$f_{\phi^\star}(\cdot \mid \underline{X}_t^{(q)}, \mathcal{D}_\tau^{1:m}) = P(Y^{(q)} \mid \underline{X}_t^{(q)}, \mathcal{D}_\tau^{1:m}).$$

We now show that the loss minimizer also recovers the joint conditional

$$\sum_{q=m+1}^N \mathbb{E}_{Y^{(q)} | \underline{X}_t^{(q)}, \mathcal{D}_\tau^{1:m}} \left[ \ell\left( f_{\phi^\star}(\cdot \mid \underline{X}_t^{(q)}, \mathcal{D}_\tau^{1:m}), Y^{(q)} \right) \right]$$

$$\stackrel{\text{change of measure}}{=} \sum_{q=m+1}^N \mathbb{E}_{Y^{(q)} | \underline{X}_t^{m+1:N}, \mathcal{D}_\tau^{1:m}} \left[ - \frac{p(Y^{(q)} \mid \underline{X}_t^{(q)}, \mathcal{D}_\tau^{1:m})}{p(Y^{(q)} \mid \underline{X}_t^{m+1:N}, \mathcal{D}_\tau^{1:m})} \log f_{\phi^\star}(Y^{(q)} \mid \underline{X}_t^{(q)}, \mathcal{D}_\tau^{1:m}) \right]$$

$$\stackrel{\text{linearity of } \mathbb{E}}{=} \mathbb{E}_{Y^{m+1:N} | \underline{X}_t^{m+1:N}, \mathcal{D}_\tau^{1:m}} \left[ - \sum_{q=m+1}^N \frac{p(Y^{(q)} \mid \underline{X}_t^{(q)}, \mathcal{D}_\tau^{1:m})}{p(Y^{(q)} \mid \underline{X}_t^{m+1:N}, \mathcal{D}_\tau^{1:m})} \log f_{\phi^\star}(Y^{(q)} \mid \underline{X}_t^{(q)}, \mathcal{D}_\tau^{1:m}) \right]$$

$$\stackrel{\text{CI assumption A.5}}{=} \mathbb{E}_{Y^{m+1:N} | \underline{X}_t^{m+1:N}, \mathcal{D}_\tau^{1:m}} \left[ - \sum_{q=m+1}^N \log f_{\phi^\star}(Y^{(q)} \mid \underline{X}_t^{(q)}, \mathcal{D}_\tau^{1:m}) \right]$$

$$\stackrel{\text{log manipulation}}{=} \mathbb{E}_{Y^{m+1:N} | \underline{X}_t^{m+1:N}, \mathcal{D}_\tau^{1:m}} \left[ - \log \prod_{q=m+1}^N f_{\phi^\star}(Y^{(q)} \mid \underline{X}_t^{(q)}, \mathcal{D}_\tau^{1:m}) \right]$$

$$\stackrel{\text{factorized predictor}}{=} \mathbb{E} \left[ - \log f_{\phi^\star}(Y^{m+1:N} \mid \underline{X}_t^{m+1:N}, \mathcal{D}_\tau^{1:m}) \right]$$

$$= H(Y^{m+1:N} \mid \underline{X}_t^{m+1:N}, \mathcal{D}_\tau^{1:m}) + \mathrm{KL}\left( p(Y^{m+1:N} \mid \underline{X}_t^{m+1:N}, \mathcal{D}_\tau^{1:m}) \,\|\, f_\phi(Y^{m+1:N} \mid \underline{X}_t^{m+1:N}, \mathcal{D}_\tau^{1:m}) \right)$$

$\square$

Part II - Proof of CMI-optimal acquisition

*Proof.* We consider the loss in equation 9, and plug $\phi^\star$ in and use Lemma A.4 for the sequence scenario.

We first rewrite the per-action expected loss in terms of the conditional entropy.

$$J(a_t; \underline{x}_t^{(q)}, \mathcal{D}_\tau^{1:m}, \phi^\star) = \mathbb{E}_{\substack{X_{a_t}^{(q)}, Y^{(q)}|\underline{x}_t^{(q)}, \\ \mathcal{D}_\tau^{1:m}, R_{a_t}=1}} \left[ \ell\big(f_{\phi^\star}(\cdot \mid \underline{x}_t^{(q)} \cup X_{a_t}^{(q)}, \mathcal{D}_\tau^{1:m}), Y^{(q)}\big) \right]$$

$$= \mathbb{E}_{\substack{X_{a_t}^{(q)}, |\underline{x}_t^{(q)}, \\ \mathcal{D}_\tau^{1:m}, R_{a_t}=1}} \left[ H(Y^{(q)} \mid X_{a_t}^{(q)}, x_t^{(q)}, \mathcal{D}_\tau^{1:m}) \right]$$

Now we use the same logic as in Theorem A.2. Plugging the per-action expected loss back into the total loss:

$$\mathcal{L}(f_\phi, \pi_\theta) = \mathbb{E}_{\substack{\tau \sim \mathcal{T} \\ \mathcal{D}_\tau^N \sim P_\tau}} \left[ \sum_{m=1}^{N-1} \mathbb{E}_{S_t^{(q)}, R} \left[ \mathbb{E}_{A_t \sim \tilde\pi_\theta(\cdot|S_t^{(q)}, \mathcal{D}_\tau^{1:m})} [J(A_t; \underline{X}_t^{(q)}, \mathcal{D}_\tau^{1:m}, \phi^\star)] \right] \right]$$

$$= \mathbb{E}_{\substack{\tau \sim \mathcal{T} \\ \mathcal{D}_\tau^N \sim P_\tau}} \left[ \sum_{m=1}^{N-1} \mathbb{E}_{S_t^{(q)}, R} \left[ \sum_{a_t^m : r_{a_t}=1} \tilde\pi_\theta(a_t^m \mid s_t^{(q)}, \mathcal{D}_\tau^{1:m}) J(a_t^m; \underline{x}_t^{(q)}, \mathcal{D}_\tau^{1:m}, \phi^\star) \right] \right]$$

Therefore, for each tuple $(s_t^{(q)}, \mathcal{D}_\tau^{1:m}, r)$, the inner summation

$$\sum_{a_t^m : r_{a_t}=1} \tilde\pi_\theta(a_t^m \mid s_t^{(q)}, \mathcal{D}_\tau^{1:m}) J(a_t^m; \underline{x}_t^{(q)}, \mathcal{D}_\tau^{1:m}, \phi^\star)$$

is linear over the simplex on $\{a_t^m : r_{a_t} = 1\}$ and is therefore minimized when we select the acquisition action $\arg\min_{a_t^m} J(a_t^m; \underline{x}_t^{(q)}, \mathcal{D}_\tau^{1:m}, \phi^\star)$. Therefore, the loss minimizer $\tilde\pi_{\theta^\star}$ places all mass on the action that minimizes the conditional entropy among the available actions, which equivalently maximizes $I(Y; X_{a_t} \mid \underline{x}_t^{(q)}, R_{a_t} = 1, \mathcal{D}_\tau^{1:m})$.

$\square$

*Remark* A.6. While Equation 5 defines the sequence loss via an expectation over a single query state for theoretical clarity, in our implementation we estimate this expectation by evaluating and averaging the loss over all valid acquisition steps $t \in \{0, \dots, T\}$ within each sampled trajectory, where $T$ is the number of available features within each sample determined by $R$. This procedure is described in Algorithm 2.

### A.4. Discussion on Greedy Acquisition

We provide additional discussion on greedy acquisition using the concept of adaptive submodularity (Golovin & Krause, 2011). Let there be $d$ available features with index set $[d] := \{1, \dots, d\}$. A full realization is $x \in \mathcal{X}^d$, where the value of feature $j$ is $x_j \in \mathcal{X}$. We aim to maximize a nonnegative utility $f : 2^{[d]} \times \mathcal{X}^d \to \mathbb{R}_{\geq 0}$, where $g(S, x)$ evaluates the utility of acquiring the subset $S \subseteq [d]$ under realization $x$.

We begin by recalling submodularity.

**Definition A.7.** A set function $g : 2^{[d]} \to \mathbb{R}$ is *submodular* if for all $A \subseteq B \subseteq [d]$ and every $j \in [d] \setminus B$,

$$g(A \cup \{j\}) - g(A) \geq g(B \cup \{j\}) - g(B).$$

Intuitively, the property of submodularity implies diminishing returns. We now recall the definition of adaptive submodularity for feature acquisition

**Definition A.8.** Let $X = (X_1, \dots, X_d) \in \mathcal{X}^d$ be random. A utility $f : 2^{[d]} \times \mathcal{X}^d \to \mathbb{R}_{\geq 0}$ is *adaptively submodular* if for all sets $S \subseteq S' \subseteq [d]$, for all indices $j \in [d] \setminus S'$, and for all partial realizations $x_S$ and $x_{S'}$, the conditional expected marginal benefit does not increase as more outcomes are observed:

$$\Delta(j \mid x_S) = \mathbb{E}_{X|x_s} [f(S \cup \{j\}, X) - f(S, X)]$$
$$\geq \mathbb{E}_{X|x_{s'}} [f(S' \cup \{j\}, X) - f(S', X)] = \Delta(j \mid x_{S'}).$$

The theoretical result in (Golovin & Krause, 2011) shows that for a fixed budget $k$, the greedy policy for a distribution that satisfies definition A.8 (and adaptive monotone) achieves an $(1 - e^{-1})$ approximation to the expected reward of the best policy, following from the result that the optimality gap shrinks by an $(1 - k^{-1})$ factor at each step. However, greedy CMI-based acquisition is not adaptively submodular for problems where features are jointly informative. For example, feature $j$ (a chest X-ray) may only be informative after a different feature has been observed due to synergistic information (an electrocardiogram), but uninformative on its own. (Norcliffe et al., 2025) provides an intuitive example using an indicator variable that determines which features are informative, and also discusses limitations of CMI if the objective is 0-1 loss minimization.

### A.5. Lookahead Variant

Given the limitations of greedy acquisition, we additional consider a variant where we include a "planning" signal by maximizing a lookahead objective.

Denote the state for the $m$-th sample in the sequence as $S_t^m = (\underline{X}_t^m, \underline{A}_{t-1}^m)$, and $A_t$ is an action sampled from $\tilde{\pi}_\theta(S_t^m, \mathcal{D}_\tau^{1:m})$. Recall the per-action expected loss is defined as the deterministic function given action $a_t$, starting query state $s_t^{(q)}$ and context $\mathcal{D}_\tau^{1:m}$

$$J(a_t; \underline{s}_t^{(q)}, \mathcal{D}_\tau^{1:m}) = \mathbb{E}_{\substack{X_{a_t}^{(q)}, Y^{(q)}|\underline{s}_t^{(q)}, \\ \mathcal{D}_\tau^{1:m}, R_{a_t}=1}}\Big[\ell\big(f_\phi(\cdot \mid \underline{s}_t^{(q)} \cup X_{a_t}^{(q)}, \mathcal{D}_\tau^{1:m}), Y^{(q)}\big)\Big],$$

where we use $\ell$ to denote the loss (ex. negative log likelihood) for evaluating the predictor.

For a general $k$-step lookahead objective ($k = 0$ recovers the greedy objective), let $\xi = \{S_{t:t+k}^{(q)}, A_{t:t+k}\}$ denote a length $k + 1$ rollout generated from $A_t \sim \tilde{\pi}_\theta(S_t^{(q)}, \mathcal{D}_\tau^{1:m})$. We define $J^{(k)}(A_{t:t+k}; \underline{s}_t^{(q)}, \mathcal{D}_\tau^{1:m})$ as the expected loss under the random trajectory $\mathbb{E}_\xi\Big[J\Big(A_{t+k}; \underline{S}_{t+k}^{(q)}, \mathcal{D}_\tau^{1:m}\Big) \mid \underline{s}_t^{(q)}\Big]$. This leads to the overall sequence prediction objective:

$$\mathcal{L}^{(k)}(f_\phi, \pi_\theta) = \mathbb{E}_{\substack{\tau \sim \mathcal{T} \\ \mathcal{D}_\tau^N \sim P_\tau}}\left[\sum_{m=1}^{N-1} \mathbb{E}_{S_t^{(q)}, R}\big[J^{(k)}(A_{t:t+k}; \underline{s}_t^{(q)}, \mathcal{D}_\tau^{1:m})\big]\right] \tag{9}$$

We optimize the above objective with a pathwise estimator using the same smooth relaxation approach. We sample $\epsilon_t \sim \text{Gumbel}(0, 1)$ from the Gumbel distribution and denote the sampled index as $\tilde{A}_t = g_\theta(\epsilon_t; S_t^{(q)}, \mathcal{D}_\tau^{1:m})$, leading to the following gradient approximation of Equation 9:

$$\nabla_\theta \mathcal{L}^{(k)}(f_\phi, \pi_\theta) = \mathbb{E}_{\substack{\tau \sim \mathcal{T} \\ \mathcal{D}_\tau^N \sim P_\tau}}\left[\sum_{m=1}^{N-1} \mathbb{E}_{S_t^{(q)}, R, \epsilon_{t:t+k-1}}\Big[\nabla_\theta J^{(k)}(\tilde{A}_{t:t+k-1}; \underline{s}_t^{(q)}, \mathcal{D}_\tau^{1:m}))\Big]\right] \tag{10}$$

where $g_\theta(\epsilon_t)$ denotes the straight-through Gumbel–Softmax estimator with temperature $\eta$, i.e., we sample $\hat{a}_t = \text{softmax}\left(\frac{\log \pi_\theta + \epsilon_t}{\eta}\right)$ and set $\tilde{A}_t$ with the hard sample in the forward pass while backpropagating through $\hat{a}_t$. We use standard Monte Carlo estimation and draw $N_{\text{MC}}$ independent noise sequences $\{\epsilon_{t:t+k-1}^{(i)}\}_{i=1}^{N_{\text{MC}}}$ and form the corresponding reparameterized action trajectories $\tilde{A}_{t:t+k-1}^{(i)} = g_\theta(\epsilon_{t:t+k-1}^{(i)}; S_t^{(q)}, \mathcal{D}_\tau^{1:m})$. The resulting pathwise gradient estimator is

$$\widehat{\nabla_\theta J^{(k)}} = \frac{1}{N_{\text{MC}}} \sum_{i=1}^{N_{\text{MC}}} \nabla_\theta J^{(k)}(\tilde{A}_{t:t+k-1}^{(i)}; \underline{s}_t^{(q)}, \mathcal{D}_\tau^{1:m}) \tag{11}$$

We demonstrate empirically as a proof of concept that using this approach, lookahead policies can be learned on a diverse task prior.

### A.6. Model Architecture and Training Details

The goal is to model the one-step predictive distributions $p(Y^{(q)} \mid \underline{X}_t^{(q)} = \underline{x}_t^{(q)}, \mathcal{D}_\tau^{1:m})$, also referred to as posterior predictive distributions (PPD). We refer to various previous works for formalizing the connection between sequence modeling and Bayesian inference (Müller et al., 2021; Nguyen & Grover, 2022; Ye & Namkoong, 2024). We leverage the insight that the sequence model for performing explicit Bayesian inference must satisfy the following inductive invariances (Nguyen & Grover, 2022; Ye & Namkoong, 2024). For this section, we denote each tuple in $\mathcal{D}_\tau^{1:m}$ as $Z^i = (X^i, R^i, Y^i)$.

A.6.1. MODEL ARCHITECTURE

**Definition A.9. Context Invariance.** A model $f_\phi$ is context invariant if for any choice of permutation function $\pi$ and $m \in [1, N-1]$, $f_\phi(Y^{m+1:N}|\underline{X}_t^{m+1:N}, Z^{1:m}) = f_\phi(Y^{m+1:N}|\underline{X}_t^{m+1:N}, Z^{\pi(1):\pi(m)})$

**Definition A.10. Target Equivariance.** A model $f_\phi$ is target equivariant if for any choice of permutation function $\pi$ and $m \in [1, N-1]$, $f_\phi(Y^{m+1:N}|\underline{X}_t^{m+1:N}, Z^{1:m}) = f_\phi(Y^{\pi(m+1):\pi(N)}|\underline{X}_t^{\pi(m+1):\pi(N)}, Z^{1:m})$

We approximate these invariances using a Transformer model (Vaswani et al., 2017) with several modifications. For each input query sample $i$, the sufficient statistics for the state $s_t^i$ are the partially observed feature values $\underline{x}_t^i \in \mathbb{R}^d$ together with the acquisition mask $\underline{a}_{t-1}^i \in \{0,1\}^d$, which records which features have been acquired so far. The state is encoded by applying the mask to the feature vector, $x_t^i \odot \underline{a}_{t-1}^i$, and concatenating this with the mask itself. Finally, we append a zero vector of length $c$ to represent the unobserved target $Y$. The resulting input representation for a query sample is

$$z_{\text{qry}}^i = \left[ \, x_t^i \odot \underline{a}_{t-1}^i, \; \underline{a}_{t-1}^i, \; \mathbf{0}^c \, \right] \in \mathbb{R}^{2d+c}.$$

For each context sample $i$, the sufficient statistics consist of the partially observed feature values $x^i \in \mathbb{R}^d$ together with the retrospective missingness mask $r^i \in \{0,1\}^d$, which indicates which features were collected in the past. We append the observed target $y^i$ to form the encoded representation.

$$z_{\text{ctx}}^i = \left[ \, x^i \odot r^i, \; r^i, \; y^i \, \right] \in \mathbb{R}^{2d+c}.$$

Next, we remove standard positional embeddings and replace the usual causal attention mask with a custom design, since causal masking does not satisfy the invariances in Definition A.9. To enable efficient computation of the autoregressive loss, we also introduce *target points* into the sequence during training.

Each input sequence for autoregressive loss computation has length $2N - m$ and is ordered as

$$\{ \, z_{\text{ctx}}^1, \ldots, z_{\text{ctx}}^m, \; z_{\text{tar}}^{m+1}, \ldots, z_{\text{tar}}^N, \; z_{\text{qry}}^{m+1}, \ldots, z_{\text{qry}}^N \, \}.$$

Each target point $z_{\text{tar}}^i$ shares the same underlying feature vector $x^i$ as its corresponding query $z_{\text{qry}}^i$, but encodes the retrospective mask $r^i$ and observed target variable $y^i$ in place of zero-padding.

The attention mask 6 enforces the following structure:

- Context points can attend freely to one another.

- Each target point can attend to all context points and all preceding target points.

- Each query point $z_{\text{qry}}^i$ can attend to context points and preceding target points, but not to other queries.

Attention Mask

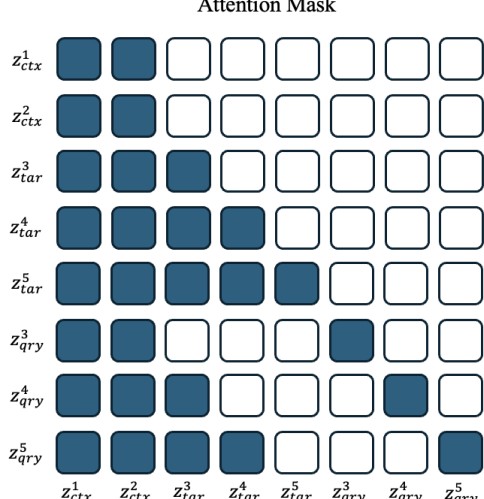

*Figure 6.* Attention mask used during training with 2 context samples and 3 query samples. Each query has a paired target sample that shares the same features but includes the observed target variable $y$ and retrospective mask $r$ instead of zero-padding.

### A.6.2. TRAINING

We provide the algorithm for pretraining the predictor in Algorithm 1 and pretraining the policy in Algorithm 2. The inference procedure is provided in Algorithm 3.

---

**Algorithm 1** Autoregressive training for sequence model $f_\phi$ given $\mathbb{P}_\mathcal{T}$

---

**Input:** Predictor $f_\phi$ **Require:** Sequence length $N$, batch size $J$

1: **for** until convergence **do**
2:     **for** each task $\mathcal{D}_\tau = \{X^{1:N}, Y^{1:N}, R^{1:N}\}$ in mini-batch **do**
3:         Initialize the set of observed indices $\underline{A}_0^{1:N}, \underline{A}_0 \subseteq [d]$ with always available feature indices
4:         **for** $t \in \{1,...,d-1\}$ **do**
5:            **for** $m \in \{1, ..., N-1\}$ **do**
6:                Predict next label using the sequence model:

$$\hat{Y}^{m+1} \sim f_\phi(\cdot \mid \underline{X}_t^{m+1}, X^{1:m}, R^{1:m}, Y^{1:m})$$

7:                Sample a random feature index $j : R_j^{m+1} = 1$ to acquire, so $\underline{A}_t^{m+1} \leftarrow \underline{A}_{t-1}^{m+1} \cup \{j\}$
8:            **end for**
9:         **end for**
10:     **end for**
11:     Compute mini-batch loss $\hat{l}_\phi$ and update parameters $\phi \leftarrow \phi - \eta \nabla_\phi \hat{l}_\phi$
12: **end for**
13: **return** trained model $\hat{f}_\phi$

---

---

**Algorithm 2** Autoregressive training for sequence model $\pi_\theta$ given $\mathbb{P}_\mathcal{T}$

---

**Input:** Policy $\pi_\theta$, predictor $f_\phi$ **Require**: Sequence length $N$, batch size $J$

1: **for until convergence do**
2:    **for** each task $\mathcal{D}_\tau = \{X^{1:N}, Y^{1:N}, R^{1:N}\}$ in mini-batch **do**
3:       Initialize the set of observed indices $\underline{A}_0^{1:N}$, $\underline{A}_0 \subseteq [d]$ with always available feature indices
4:       **for** $t \in \{1, ..., d-1\}$ **do**
5:          **for** $m \in \{1, ..., N-1\}$ **do**
6:             Given the state $S_t^{m+1} = (\underline{X}_t^{m+1}, \underline{A}_{t-1}^{m+1})$, output action distribution using policy:

$$\hat{A}^{m+1} \sim \pi_\theta(\cdot \mid S_t^{m+1}, X^{1:m}, R^{1:m}, Y^{1:m})$$

7:             Approximate argmax using straight through gumbel-softmax: $\tilde{A}^{m+1}$
8:             Compute and accumulate one-step loss using the predictor:

$$\ell\big(f_\phi(\cdot \mid \underline{X}_t^{m+1} \cup X_{\tilde{A}}, \, X^{1:m}, R^{1:m}, Y^{1:m}), Y^{m+1}\big)$$

9:             Sample a random feature index $j : R_j^{m+1} = 1$ to acquire, so $\underline{A}_t^{m+1} \leftarrow \underline{A}_{t-1}^{m+1} \cup \{j\}$
10:          **end for**
11:       **end for**
12:    **end for**
13:    Compute mini-batch loss $\hat{l}_\theta$ and update parameters $\theta \leftarrow \theta - \eta \nabla_\theta \hat{l}_\theta$
14: **end for**
15: **return** trained model $\hat{\pi}_\theta$

---

**Algorithm 3** Test-time inference procedure for solving an AFA task $\tau$

---

**Require:** Pretrained sequence models $f_\phi$, $\pi_\theta$   **Input:** Samples from $\mathcal{D}_\tau = \{X^{1:m}, R^{1:m}, Y^{1:m}\}$, and query samples $\underline{X}_0^{m+1:N}$ feature budget $k \le d$

1: **for** $t \in \{1, \ldots, k\}$ **do**
2:    **for** $q \in \{m+1, \ldots, N\}$ **do**
3:       Compute $\pi_\theta(A_t^{(q)} \mid \underline{X}_t^{(q)}, \underline{A}_{t-1}^{(q)}, \mathcal{D}_\tau)$ and select action

$$a_t^i = \arg\max_a \; \pi_\theta(a \mid \underline{X}_t^{(q)}, \underline{A}_{t-1}^{(q)}, \mathcal{D}_\tau)$$

4:       Update $\underline{X}_{t+1}^{(q)} \leftarrow \underline{X}_t^{(q)} \cup X_{a_t}$ with chosen action
5:    **end for**
6: **end for**
7: **Return:** Predictions for all test samples in task $\tau$

$$\hat{Y}^i \sim f_\phi\big(\cdot \mid \underline{X}_k^i, \mathcal{D}_\tau\big), \quad \forall i \in \{m+1, \ldots, N\}$$

---

## A.7. Experiment Details

### A.7.1. DATASETS

For each dataset, we construct two disjoint splits: a training set of size $n_{\text{train}}$ and a test pool of size $n_{\text{test}}$. For each dataset, we specify three components:

- **Baseline features** ($X_0$): features that are always observed at the start of an acquisition trajectory.

- **Acquirable features** ($X_m$): candidate features available for sequential acquisition.

- **Label space** ($Y$): the target variable or class labels used for evaluation.

*Table 1.* Label prevalences (%) for final evaluation datasets.

| Dataset | % Positive |
|---|---|
| MiniBooNE | 72.16 |
| MIMIC-LOS | 44.90 |
| MIMIC-Readmission | 16.52 |
| MIMIC-Mortality | 8.67 |

1. **Metabric** ($n_{\text{train}} = 1,000$, $n_{\text{test}} = 898$):

$$X_m = \{\texttt{ccnb1, cdk1, e2f2, e2f7, stat5b, notch1,}$$
$$\texttt{rbpj, bcl2, egfr, erbb2, erbb3}\}.$$

$$Y \in \{\texttt{Luminal A, Luminal B, HER2-enriched,}$$
$$\texttt{Basal-like, Normal-like, Claudin-low}\}.$$

2. **MiniBooNE** ($n_{\text{train}} = 5,000$, $n_{\text{test}} = 10,000$):

$$X_m = \{\texttt{Feature 1, Feature 17, Feature 23, Feature 32, Feature 3,}$$
$$\texttt{Feature 27, Feature 12, Feature 4, Feature 25, Feature 2}\}.$$

$$Y \in \{\texttt{0,1}\}.$$

3. **MNIST** ($n_{\text{train}} = 30,000$, $n_{\text{test}} = 30,000$):

$$X_m = \{\texttt{block\_0\_4, block\_1\_6, block\_2\_2, block\_3\_5, block\_0\_3,}$$
$$\texttt{block\_0\_2, block\_5\_2, block\_4\_6, block\_5\_0, block\_4\_2,}$$
$$\texttt{block\_4\_3, block\_5\_6, block\_6\_3, block\_3\_3, block\_1\_3,}$$
$$\texttt{block\_5\_1, block\_4\_4, block\_3\_2, block\_5\_5, block\_2\_4}\}.$$

$$Y \in \{0, 1, 2, 3, 4, 5, 6, 7, 8, 9\}.$$

4. **MIMIC-IV** (Johnson et al., 2023) ($n_{\text{train}} = 5,000$, $n_{\text{test}} = 3,000$):

$$X_0 = \{\texttt{Age, Gender, ICU, Diabetes, CPD}\}, \quad X_m = \{\texttt{Hemoglobin, Platelet, RBC, WBC,}$$
$$\texttt{Bicarbonate, BUN, Calcium, Chloride, Creatinine, Glucose, RDW, INR,}$$
$$\texttt{PT, lymphocytes, monocytes}\}.$$

$$Y_{\text{Length of stay}} \in \{\texttt{0,1}\},$$
$$Y_{\text{Mortality}} \in \{\texttt{0,1}\},$$
$$Y_{\text{Readmission}} \in \{\texttt{0,1}\}.$$

**Preprocessing** We define three binary classification tasks on **MIMIC-IV**. For each unique patient, we retain a single admission and set the prediction time to 48 hours after admission. Patients with an admission shorter than 48 hours are excluded. For each feature, we use the most recent measurement recorded before the prediction time; if no measurement is available from admission up to prediction time, the feature is treated as missing. To ensure that all ground-truth measurements are available for evaluation, we also exclude patients with missing values in any of the selected features in $X_m$. The tasks are defined as follows:

- **Length of stay (LOS):** whether the hospital stay extends at least 7 days beyond the prediction time.

- **Mortality:** whether the patient dies during the same hospital admission.

- **Readmission:** whether the patient is readmitted to the hospital within 30 days of discharge.

The tasks are generated using MEDS to facilitate reproducibility (McDermott et al., 2025).

A.7.2. PRETRAINING TASK PRIOR

Here we describe the synthetic task prior used for pretraining our **L2M** models.

**GP.** We define a Gaussian process task prior with an RBF kernel to generate synthetic regression tasks. For each sampled task, we randomly select a subset of the input dimensions to be informative, while the remaining dimensions are treated as noise features. The kernel is parameterized with batch-specific lengthscales and output scales: lengthscales are drawn uniformly from the interval $[0.1, 5.0]$ for each dimension, and output scales are drawn uniformly from $[0.5, 2.0]$. Non-informative features are assigned a large lengthscale, effectively removing their contribution. An observation noise term $\sigma_\epsilon^2 I$ with $\sigma_\epsilon = 2 \times 10^{-2}$ is added for numerical stability.

**BNN.** We define a Bayesian neural network (BNN) task prior for classification tasks that generates synthetic labeling functions over feature inputs. For each sampled task, we proceed as follows:

1. **Selection of informative features.** For each batch, we randomly select a subset of features between $[\texttt{min\_feats}, \texttt{max\_feats}]$. Data points are grouped into 1–3 clusters by generating cluster centers sampled from a Gaussian distribution. Each datapoint $x \in \mathbb{R}^d$ is assigned a cluster label via its closest cluster center according to Euclidean distance. Then each cluster is assigned its own subset of informative features. Features not selected are masked out and do not influence the label.

2. **Random BNN weights.** A two-layer feedforward neural network with hidden dimension $H = 8$ and $\texttt{tanh}$ nonlinearity is constructed. Weights and biases are drawn from Gaussian distributions and scaled by random importance weights and scale factors sampled uniformly from given ranges. The masked input features are passed through the random network, producing output logits.

3. **Task-specific adjustments.** Logits are rescaled by a random temperature parameter sampled from a uniform range. For binary tasks, a random bias shift is applied to match a target label prevalence $p \in [0.05, 0.95]$.

4. **Label generation.** The final logits are passed through a sigmoid to produce probabilities, from which labels $Y$ are sampled as Bernoulli (for binary classification) or categorical (for multi-class) random variables.

This procedure defines a flexible family of tasks where both the informative feature subsets and the underlying labeling functions vary across tasks, simulating heterogeneity in feature importance for AFA.

**BNN-Plan.** We additional define a Bayesian neural network (BNN) task prior for classification tasks for evaluating planning capabilities of the 1-step lookahead variant:

1. **Selection of informative features.** For each batch, we randomly select a subset of features between $[\texttt{min\_feats}, \texttt{max\_feats}]$. Features not selected are masked out and do not influence the label.

2. **Random BNN weights.** We use the same network sampling procedure as **BNN** to generate a main effect term.

3. **Additional interactions.** Out of the informative features, we randomly select pairs of features to form pairwise interactions. If the element-wise product of a pair $x_i x_j$ is above a threshold $\tau$, the label receives additional signal proportional to a randomly weighted sum $w x_i x_j$ of the products. This signal is added to the main effect term.

4. **Label generation.** After the same task-specific adjustments as **BNN** (logits scaling and shift), the final logits are passed through a sigmoid to produce probabilities, from which labels $Y$ are sampled as Bernoulli (for binary classification) or categorical (for multi-class) random variables.

A.7.3. ADDITIONAL TRAINING DETAILS

At each training step, we sample sequences of length $N$ from the pretraining pool. Feature values are normalized using the mean and variance of each feature within the task sequence. Missingness is introduced either by randomly dropping features (MCAR) or by sampling feature-specific missingness mechanisms from the BNN prior (MAR) that depend only on the baseline covariates $X_0$. The missingness rates vary by feature, and we set the maximum probability of missingness $p(R_j = 0|X_0) \leq 0.5$. A summary of the experimental design is provided in Table 2.

*Table 2.* Experimental setup across datasets. Each training step samples sequences of length $N$ from the pretraining pool. Feature values are normalized within each sequence.

| Dataset | Sequence length $N$ | Missingness | Task |
|---|---|---|---|
| **GP** | 500 | MCAR | Regression |
| **MiniBooNE** | 1000 | MCAR | Binary Classification |
| **MNIST** | 1000 | MCAR | Binary Classification |
| **Metabric** | 500 | MCAR | Multi-class Classification |
| **MIMIC-IV** | 1000 | MAR | Binary Classification |

### A.7.4. COMPUTE DETAILS

All experiments were run on a server with 4 NVIDIA H100 NVL GPUs, 2 Intel(R) Xeon(R) Platinum 8480+ CPUs (56 cores each) with 2Tb of memory.

### A.7.5. RUNTIMES

The pretraining procedure for both the predictor and policy using the GP prior (sequence length of 500, 10 features, 100000 training steps) takes approximately 10 hours total on a single GPU.

### A.7.6. HYPERPARAMETERS

For our **L2M** model, we use the same hyperparameter configurations for all our experiments as shown in 3. Unless specified otherwise, we use the greedy variant with the following hyperparameters. For pretraining the predictor, all models are trained for 100000 steps with a batch size of 8 tasks (with the exception of **MNIST**, which was trained for 50000 steps). The predictor is trained with the Adam optimizer with the default optimizer parameters and with linear decay. We evaluate validation loss over a fixed set of tasks at every 500 steps and save the model with the best validation loss.

For the policy, we use a fixed temperature of 0.1 and a batch size of 8 for a total of 50000 training steps, with no learning rate decay. The transformer backbone and predictor weights are also jointly updated, with a lower learning rate of $1 \times 10^{-5}$.

| Hyperparameter | Value |
|---|---|
| Hidden Layer Size | 512 |
| Model Dimension | 256 |
| Number of Layers | 6 |
| Attention Heads | 4 |
| Embedding Depth | 4 |
| Dropout | 0 |
| Predictor Learning Rate | $1 \times 10^{-4}$ |
| Policy Learning Rate | $1 \times 10^{-4}$ |
| Warmup steps | 500 |

*Table 3.* Transformer Model Hyperparameters

**Planning experiments** For evaluating the 1-step lookahead variant, we use a fixed temperature of 0.5 and freeze the encoder and predictor when training the policy. We use $N_{\mathrm{MC}} = 4$ for computing the gradients. For a fair comparison, we train the greedy variant under the same procedure (including the same frozen components and gradient-estimation budget).

### A.7.7. BASELINES

We describe the baselines used in our experiments, noting several modifications made to improve computational tractability when evaluating across a large number of tasks.

**MLP (Random).** For each evaluation task, we train a two-layer multilayer perceptron (MLP) with hidden dimension 128. The model is trained on randomly selected feature subsets to predict the target label, using a batch size of 64 for 300 epochs. At test time, acquisition actions are chosen uniformly at random, and the MLP is used to make predictions. This task-specific

MLP serves as the predictor model for the remaining baselines.

**GDFS** (Covert et al., 2023). The policy network is also a two-layer MLP with hidden dimension 128. In contrast to the original paper, which trains the selector policy using Gumbel-softmax with a temperature decay schedule, we train using the straight-through Gumbel-softmax estimator with a fixed temperature of 0.5.

**DIME** (Gadgil et al., 2024). The reward predictor is also a two-layer MLP with hidden dimension 128. We train the reward predictor using random acquisitions, rather than the $\epsilon$-greedy acquisition strategy with decay as described in the original paper.

**DQN** (Janisch et al., 2019) We adopt the Q-learning framework, where the action-value function $Q(s_t, a)$ estimates the expected return from state $s_t$ after taking action $a$. The optimal acquisition actions are selected by taking the action with the largest Q-value estimate $Q(s_t, a)$.

We consider a dueling network architecture (Wang et al., 2016). The dueling network consists of two MLPs with two hidden layers of dimension 128: one head outputs $V(s_t)$, and the other outputs $A(s_t, a)$ for all actions $a$. The final Q-value estimate is computed as

$$Q(s_t, a) = V(s_t) + \Big(A(s_t, a) - \tfrac{1}{|\mathcal{A}|} \sum_{a'} A(s_t, a')\Big),$$

Because the final log likelihood is identical across complete trajectories, we apply a strong discount factor to prioritize early acquisitions that reduce predictive loss. The one-step temporal-difference (TD) target is defined as

$$y_t = r_t + \gamma \max_{a'} Q_{\theta^-}(s_{t+1}, a'),$$

where $r_t$ is the immediate reward, $Q_{\theta^-}$ denotes the target Q-network, and $\gamma = 0.9$ is the discount factor. We train for 200 episodes using an experience buffer of size 10,000, with samples collected via an $\epsilon$-greedy strategy. Training updates use mini-batches of 128 samples, and the target Q-network is synchronized every 4 episodes.

### A.7.8. SOURCE CODE

https://github.com/reAIM-Lab/Learning-To-Measure

## A.8. Additional Results

### A.8.1. UNCERTAINTY QUANTIFICATION

We perform evaluations of uncertainty estimation for the classification tasks. We first evaluate the ability for **L2M** to recover the ground true probabilities in a set of evaluation tasks randomly sampled from the same semi-synthetic BNN task prior used during training. To evaluate uncertainty quantification for classification, we compute the KL divergence and brier score (MSE) between the predicted probabilities and ground truth probabilities. We also additionally show the AUROC to assess the ability for the model to rank samples.

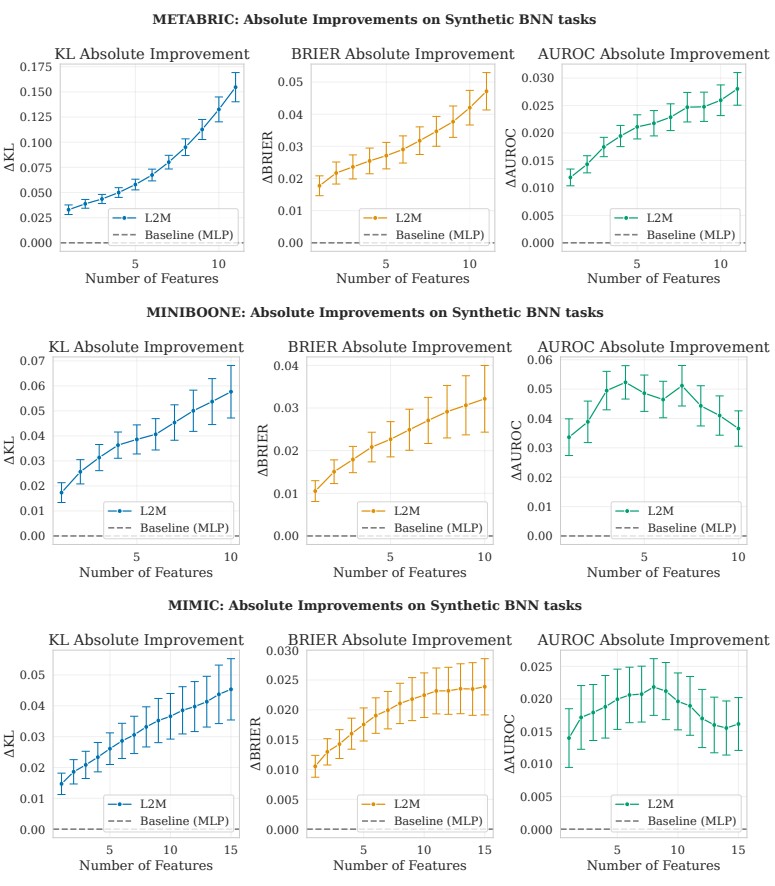

*Figure 7.* We identify a similar pattern where the quality of uncertainty quantification is better than the task-specific MLP, and the gains are larger as we acquire more features.

Next, we evaluate **L2M** on classification using semi-synthetic tasks built from real labels that were unseen during training. Because these tasks provide hard (binary) labels rather than ground-truth probabilities, we report negative log-likelihood (binary cross-entropy) and Brier score to assess uncertainty. Accordingly, we do not use KL divergence in this setting.

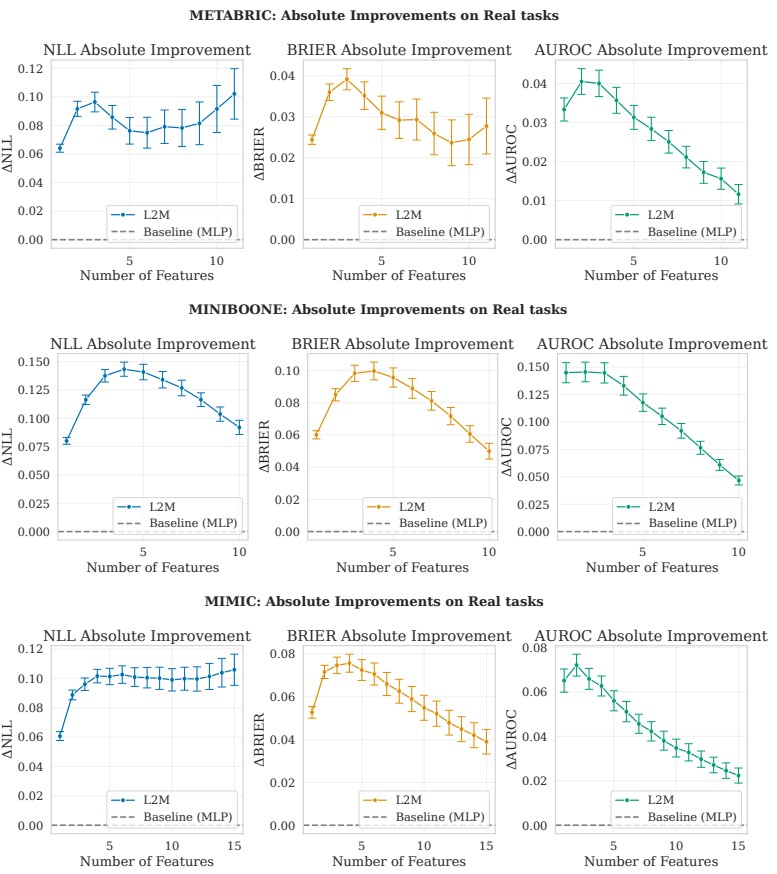

*Figure 8.* The performance on real tasks is mixed, and dependent on various factors such as whether the pretraining BNN tasks are closely aligned to the real unseen tasks.

Finally, we show the analogous result for the GP regression tasks using the same set of tasks in Figure 3. We also use mean squared error (MSE) of the predicted mean to assess the quality of **L2M** estimates.

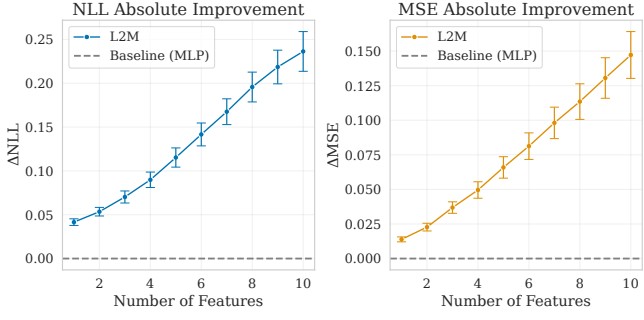

*Figure 9.* NLL and MSE improvements across acquisition step. The **L2M** approach shows progressively increasing gains as more features are acquired in the trajectory.

### A.8.2. AFA POLICY EVALUATION

**Performance**  While the main text focuses on the policy's ability to reduce uncertainty, we report additional performance metrics for AFA: MSE for regression tasks and AUROC for classification tasks. We evaluate on the same set of tasks and samples as in Figure 4.

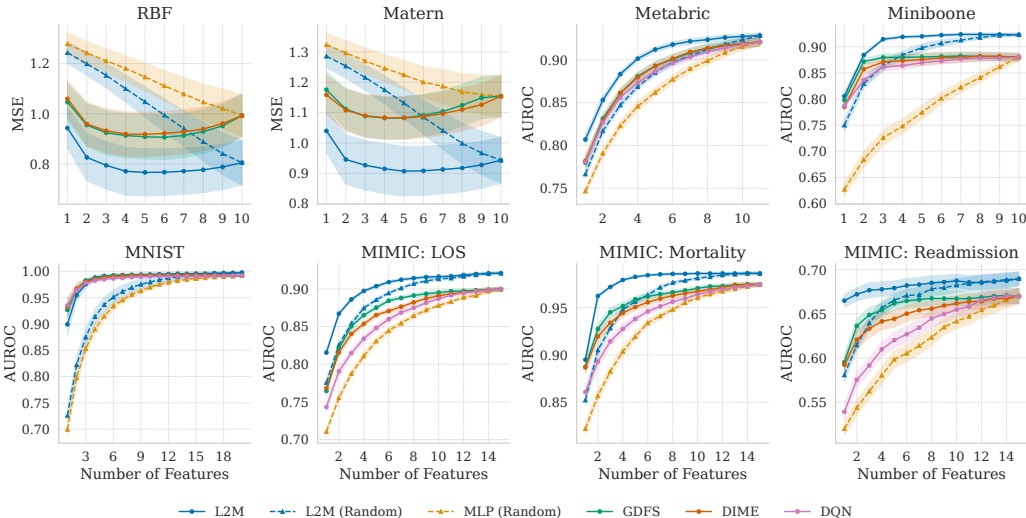

*Figure 10.* We show that **L2M** maintains improved performance under this evaluation, achieving lower MSE on regression tasks and higher AUROC on classification tasks.

**Context Length and Missingness Rates**  We demonstrate that the **L2M** provides the largest benefits in scenarios where there is lower number of context samples, and higher rates of missingness.

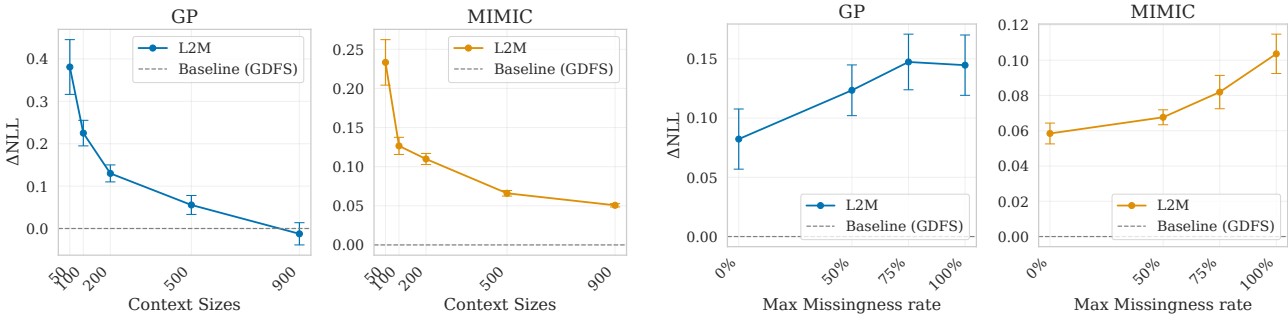

*(a) NLL improvement across context lengths*     *(b) NLL improvement across missingness rates*

*Figure 11.* Average improvement in log loss (y-axis) for the GP tasks and MIMIC-LOS for a fixed set of evaluation tasks while varying the number of context samples, and levels of retrospective missingness in each task (x-axis). Task-specific improvements are averaged over 100 (GP) and 50 (MIMIC) tasks. **L2M** is robust to settings with fewer shots (labeled samples) and with higher rates of feature missingness. **L2M** also achieves length generalization for the GP task, as it is only trained on sequences of 500 samples.

**Acquisition on simulated tasks**  We evaluate our model using tasks sampled from our synthetic pretraining prior where underlying feature informativeness is known apriori. We plot the precision and recall achieved by the acquisition method at each budget $k$, as well as the log loss metrics. We use the **MIMIC-IV** test distribution for the covariates $X$, and sample synthetic labels $Y$ using the synthetic prior. We sample 50 test tasks and evaluate with different context lengths.

**Fully synthetic variant**  We further evaluate the fully synthetic variant of **L2M**. This setup trains on a much more diverse task prior, where each task is defined by labels sampled from **BNN** and covariates sampled from multivariate Gaussian distributions with randomly sampled means and covariances. Across tasks, we allow the feature dimensionality to vary. For

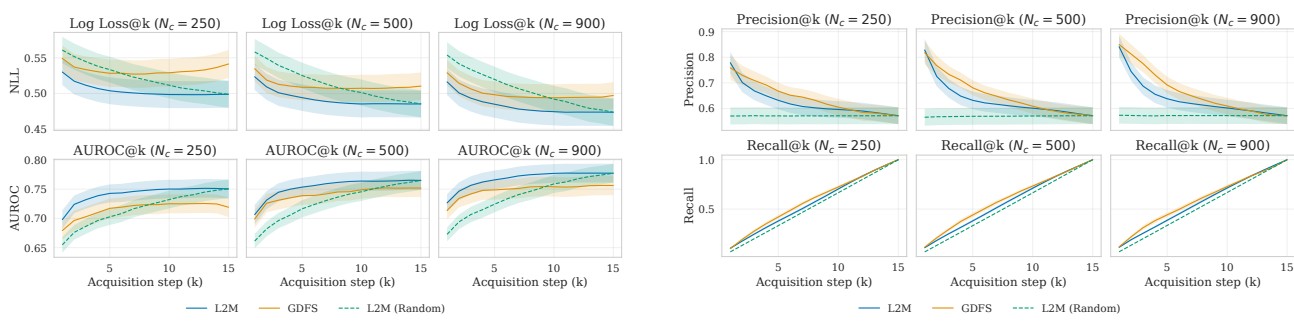

Log Loss and AUROC        Precision and Recall

*Figure 12.* While the meta-learning approach demonstrates stronger uncertainty quantification (log loss) and discrimination (AUROC) performance at each decision budget, the task-specific approach is slightly more precise when identifying informative features. The task-specific model may be learning a sharper ranking over the features, while the meta-learning approach maintains uncertainty over the optimal acquisition actions.

evaluation, we sample only simulated tasks with at least 5 features available for acquisition, while leaving the total number of features per task flexible. We note that this setting is particularly challenging for meta-learning, as the task prior must encompass heterogeneous feature distributions with varying dimensionalities. We sample 100 test tasks and evaluate with different context lengths.

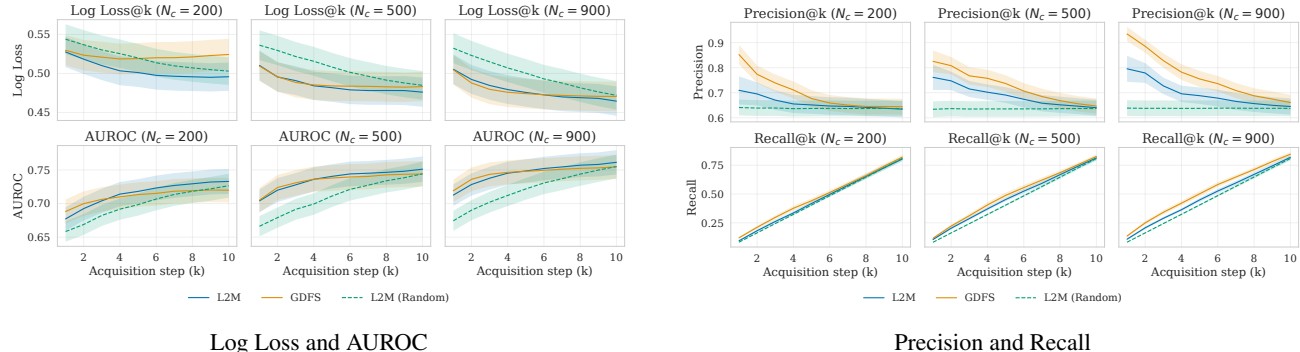

Log Loss and AUROC        Precision and Recall

*Figure 13.* **L2M** is able to match the performance of the task-specific baseline even in this challenging scenario. However, we find similarly that the task-specific approach is slightly more precise when identifying informative features.

We also evaluate the fully synthetic **L2M** model on real-world binary classification tasks. This is a particularly challenging setup, since the model never observes the real-world task's covariate distribution or label mechanism during pretraining. We denote this baseline as **L2M-Syn**, as the model is trained on fully synthetic data and applied in a zero-shot fashion to an out-of-distribution setting. We find that **L2M-Syn** acquires features that outperform random selection on the real-world tasks, but its performance degrades when compared to an **L2M** model trained directly on the real-world covariate distributions. We additionally run an ablation where, for the fully synthetic variant, we perform task-specific fine-tuning using training samples from the target (real) task. Using a synthetic pretrained model as a starting point for continual training on real data is a known strategy (Garg et al., 2025). While this approach can substantially improve performance, it comes at the cost of the intended efficiency benefits of in-context learning. Ideally, with a diverse prior specification and increased scale, the model can perform zero-shot inference and adaptation to unseen tasks without additional gradient-based training.

**Benefit of Adaptivity**  Real-world tasks such as LOS and mortality prediction may depend on different mechanisms across patients. We provide an ablation where we use the same **L2M** model but instead of the per-instance optimal action predicted by the model, we take the most frequently selected action across the entire test set at each step (majority vote). We find that the instance-wise adaptive feature selection is beneficial for some tasks, but other simpler tasks do not require granular acquisition.

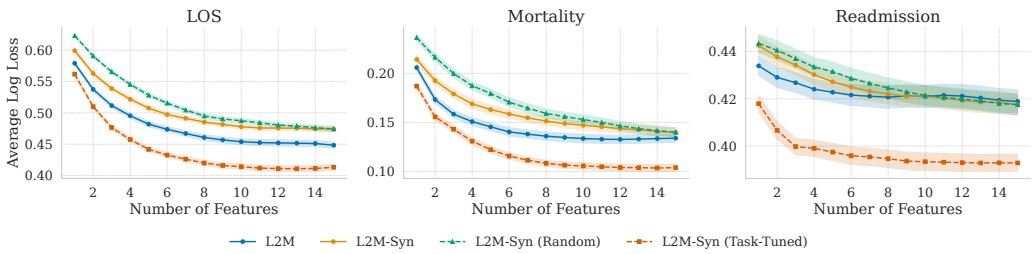

*Figure 14.* The fully synthetic variant is less effective on real-world tasks, largely because the covariate distribution it was trained on differs substantially from the real-world data. Continual training on the real-world task distribution improves performance.

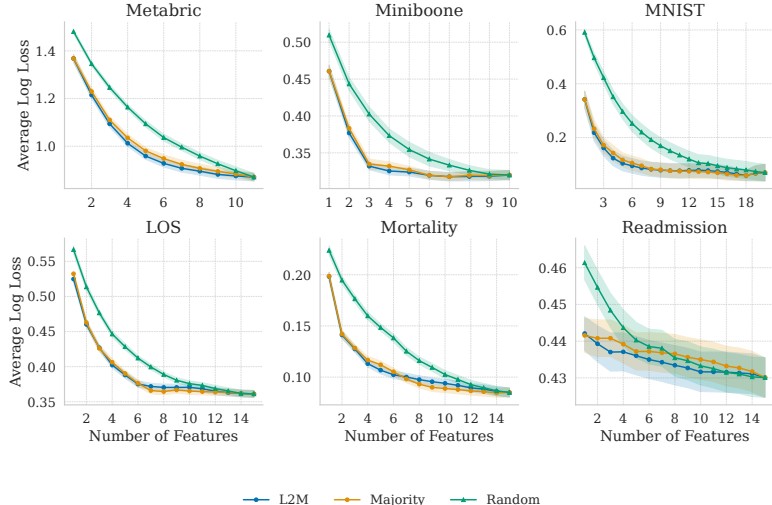

*Figure 15.* Simpler tasks such as Miniboone do not benefit from instance-wise adaptive selection. We also note that it is difficult to demonstrate the benefit of per-instance adaptivity in small-scale MIMIC experiments.

**Real missingness patterns** We evaluate **L2M** when the context set exhibits real-world missingness patterns not seen during pretraining. These results should be interpreted with caution. The model was pretrained only on complete cases with synthetically injected missingness, so the evaluation introduces a distribution shift in both patient characteristics and missingness mechanisms. In addition, the query patients are complete cases while the context patients may be partially observed, creating a context–query shift. We therefore present these results as a stress test, not a definitive measure of deployment performance.

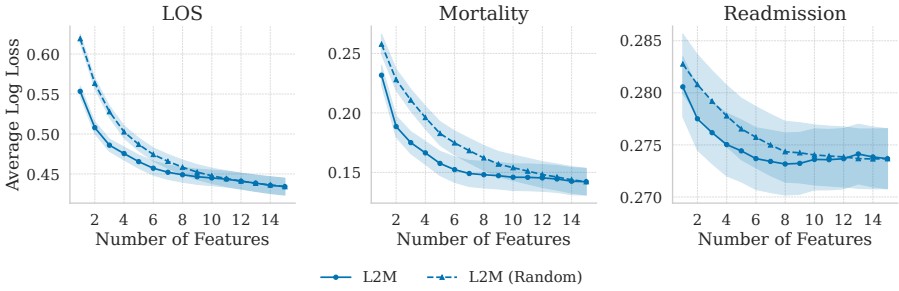

*Figure 16.* LTM is able to learn policies that improve on random acquisition on unseen natural missingness patterns in real-world data.

### A.8.3. GRADIENT ESTIMATION EVALUATION

We show that pathwise gradient estimation substantially improves sample efficiency for estimating policy gradients. We first construct a high-accuracy reference gradient using REINFORCE. To construct the evaluation, we use the pretrained

**L2M** predictor model but keep the policy randomly initialized, ensuring equal probability mass (support) over all available actions. We sample evaluation tasks and for each task, draw 500 context samples and 500 query samples with randomly masked features. For each partially observed query, we estimate the REINFORCE policy gradient by averaging over 1000 sampled acquisition trajectories, yielding a reference gradient with respect to the policy parameters $\theta$.

For the same evaluation samples, we compute the Monte-Carlo estimate of the policy gradient using varying number of trajectories, using both the REINFORCE estimator and our pathwise estimator at varying temperatures. We report the $l_2$ error/norm of the gradient with respect to the reference in Figure 17.

We find that the pathwise estimator, while biased due to the straight-through relaxation, is able to estimate the gradient with low error due to lower variance. We find that the gains are larger for the **BNN-Plan** which draws from a more diverse distribution of tasks. We also find that pathwise estimation with lower temperatures seem to struggle possibly due to higher variance.

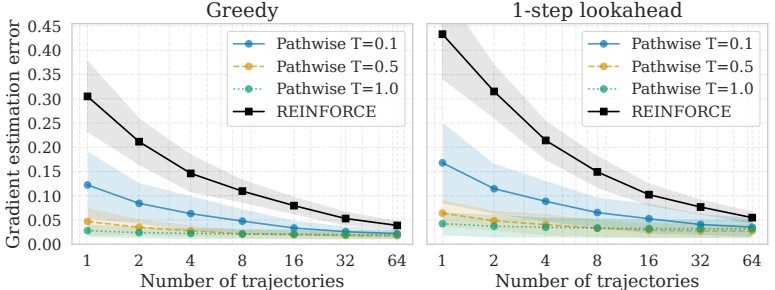

*(a)* Mean and Standard deviation from 50 fully synthetic evaluation tasks from **BNN-Plan**

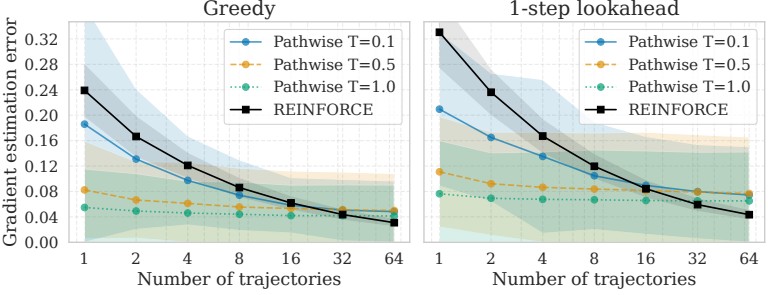

*(b)* Mean and Standard deviation from 50 semi-synthetic tasks from **MIMIC-IV**

*Figure 17.* Across 50 sampled tasks, we demonstrate that pathwise estimation of gradients using our approximation leads to low error. We evaluate compared to an unbiased reference gradient estimated using REINFORCE with 1000 trajectories per sample.

### A.8.4. META-LEARNING ABLATIONS

We provide additional results investigating alternative meta-learning strategies. One common method uses a conditional neural process (CNP) -like (Garnelo et al., 2018) approach to learn compact task representations of variable length context, and use these embeddings for downstream decision-making (Rakelly et al., 2019; Wang et al., 2024). To investigate whether this is an effective approach, we use our same transformer architecture, but perform multihead-attention pooling to learn a task-level embedding instead of full context attention. We use the same pretraining priors, and greedy differentiable policy learning. Additionally, we perform MAML-like (Finn et al., 2017) inner gradient updates for the policy and predictor head for each unseen query task encountered during inference, based on the learned CNP task representation. In other words, the model weights are updated using gradients at test-time on the set of context samples for a given query tasks.

The results in 18 show that a CNP-like approach leads to degradation in the quality and stability of uncertainty estimates. However, when pretraining on a small, fixed task family (**MNIST**) the task embedding leads to slightly improved performance and data efficiency during earlier acquisition steps. We hypothesize that this is due to the compact representation acting as an information bottleneck and regularizing the model, simplifying greedy policy learning.

We find that gradient-based adaptation at inference offers only modest improvements and rapidly overfits, indicating that the

compact task representation acts as a bottleneck that the test-time adaptation cannot overcome. The main exception is the **MIMIC-IV** mortality task, where we see consistent gains. We hypothesize this is because very low positive prevalence is underrepresented in the pretraining prior, so adaptation corrects this mismatch.

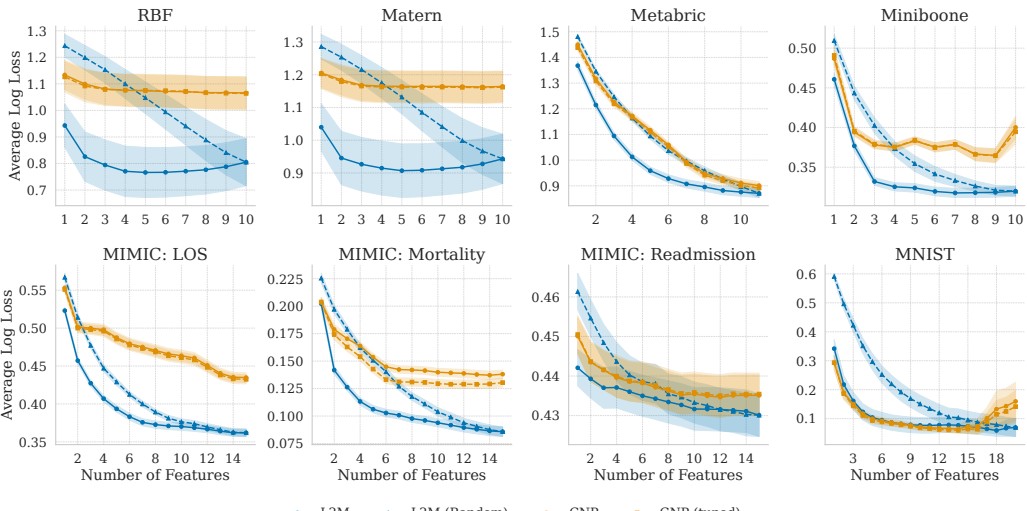

*Figure 18.* A CNP-like approach that learns a compact task representation via attention pooling leads to less stable uncertainty estimates, especially when pretrained on a diverse task prior.

A.8.5. POLICY VISUALIZATIONS

We demonstrate how our sequence modeling approach is able to learn task-specific policies. We visualize the selected actions by the greedy policy in example evaluation tasks. We show both synthetic tasks sampled from the **BNN** prior, and tasks using real labels from **MIMIC-IV**.

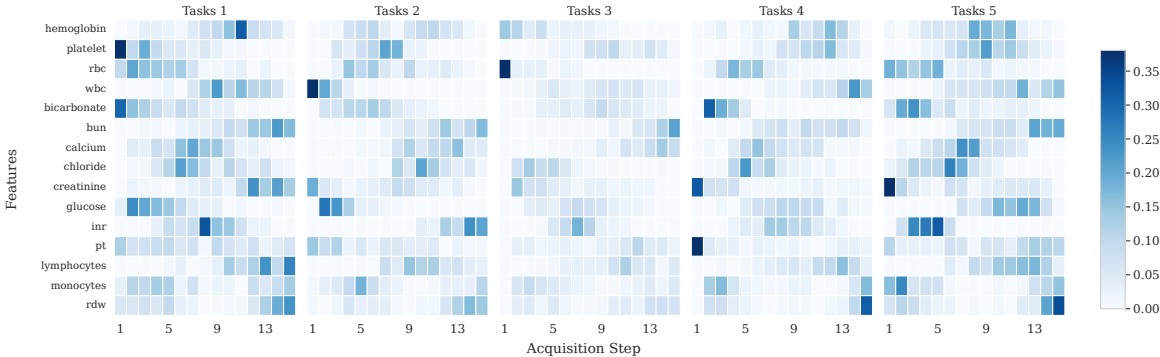

*Figure 19.* **MIMIC-IV** Dataset: Example feature acquisition patterns on semi-synthetic evaluation tasks constructed from the BNN prior. Each task contains 500 query samples, and the heatmap intensity indicates the fraction of query samples for which a given feature is selected at each acquisition step.

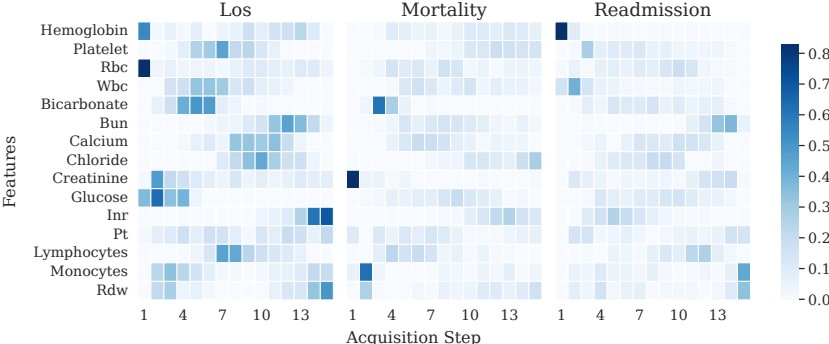

*Figure 20.* **MIMIC-IV** Dataset: Example feature acquisition for a set of 500 query samples on the semi-synthetic tasks with real labels.

