# OpenReview forum: "Learning-To-Measure: In-Context Active Feature Acquisition"
_ICML.cc/2026/Conference — ICML 2026 regular_

### Official Review · Reviewer_huwr · 2026-03-02

**Soundness:** 2
**Presentation:** 1
**Significance:** 3
**Originality:** 2
**Overall Recommendation:** 4
**Confidence:** 1

**Summary:**

This paper proposes a meta-learning method for active feature acquisition that leverages labeled data across multiple tasks with task-specific missingness. The proposed method uses a shared autoregressive Transformer model for modeling both the predictor and policy functions. The predictor estimates the class posterior given a query input with incomplete features, and the policy selects which features to observe.
After training, the policy can select appropriate features that maximize the conditional mutual information given a query and the context (i.e., historical labeled data) for unseen target tasks. The experiments with synthetic and real-world datasets show the effectiveness of the proposed approach.

**Compliance With Llm Reviewing Policy:**

Affirmed.

**Final Justification:**

Since the authors have promised to improve the presentation of the paper, which was my main concern, I will raise my rating by one point. However, since I am not an expert in AFA, I may not be able to properly assess the technical novelty and soundness. Therefore, I would ask the meta-reviewers to take this into account when making their decision.

**Key Questions For Authors:**

- Would the proposed method still work even when the tasks used for training are less related to the task used for inference?
- In Section 1, it is stated that existing methods produce bias when retrospective missingness is present. Does the proposed method resolve this? If yes, how does it do so?

**Limitations:**

Yes.

**Strengths And Weaknesses:**

Strengths
- Meta-learning for active feature acquisition (AFA) is interesting and important in some real-world applications, such as medical care. However, I am not an expert in AFA and thus cannot evaluate the novelty of this paper.
- Experiments with synthetic and real-world datasets show the effectiveness of the proposed approach.

Weakness
- The clarity of the presentation might be improved. For example, in Section 4.1, the problem setting description is unclear because it is not specified whether query samples have labels (Based on Eq. (4) and Algorithm 1, it seems that labels are likely available). Figure 4 and the description in Section 4.2 alone make it difficult to understand the model's structure. However, these concerns may also stem from my lack of familiarity with this field.

Minors
- Figure 4 is described earlier in the text than Figure 3, but it is easier to understand if explained in the order of the figure numbers.

---

> ### Author Rebuttal · Authors · 2026-03-31
>
> We thank the reviewer for the careful reading and thoughtful questions, particularly regarding task generalization and handling of retrospective missingness. We address the concerns and questions below:
>
> **[W1] Label availability during training vs. inference.** We thank the reviewer for identifying this ambiguity and will provide additional clarification in the revised manuscript. Regarding the query labels, Section 4.2 (Eq. (4) and Algorithm 1) describes the training procedure, where labels are available and used to optimize the objective. Specifically, in Equation 4,$Z^{1:m} = \\{X^{1:m}, R^{1:m}, Y^{1:m}\\}$ denotes samples which include observed true labels). However, during test time (in Section 4.1), we do not observe the label and we need to infer it from the data. The goal is to reduce uncertainty about these unknown labels efficiently by acquiring features.
>
> *Action* We recognize that the transition from the oracle problem formulation to learning approximations from data is not sufficiently delineated in the current manuscript, and we will make this distinction explicit by introducing subsection 4.1 as the test-time feature acquisition and unobserved label prediction problem, and subsection 4.2 as our proposed training methodology.
>
> **[W2] Model structure.**
> Our model is a standard transformer with minor modifications such as permutation invariance, and full architecture details are provided in Appendix A.6. As our aim is to present the conceptual framework as a general methodology, the specific implementation choices are flexible and serve as a proof of concept. We opted to keep detailed architecture specifics in the appendix accordingly.
>
> *Action:* To better emphasize our algorithmic contribution of enabling in-context learning for AFA, we will revise Figure 2 and expand Section 4.2 to clarify the training algorithm and key design choices, including input sequence structure and how the modified attention mask ensures invariance to context sample ordering.
>
> **[W3]** Thank you for pointing this out. We will swap the figure positions.
>
> **[Q1] Out-of-distribution tasks** We thank the reviewer for this important question. Although theoretical guarantees are limited to tasks drawn from distributions observed during training [1], our experiments provide empirical evidence that L2M generalizes to out-of-distribution tasks not explicitly seen during training. Specifically, for simulated tasks, the pretraining tasks are sampled from GPs with RGF kernels and evaluated on GPs with Matérn kernels. For MIMIC, we trained the model with synthetic BNN labels, and evaluated our approach on mortality, length of stay, and readmission, which has not been seen in pretraining. We show that L2M can select an appropriate learned mechanism within its prior to achieve strong performance. However, this also underscores the importance of diverse and broad task generation during pretraining to maximize the practical generalizability of the learned prior.
>
> **[Q2] Bias under missingness.** Our proposed method is unbiased when missingness is present, and we achieve this by sampling acquisition actions in the training data from only available features (using a blocked policy during training as described in Definition 3.1). We note that existing methods that rely on conditional mean imputation (replacing all missing values in the training data with point estimates of $\mathbb{E}[X_j \mid X_0, Y]$ when $R_j = 0$) are biased as it doesn’t take into account the uncertainty and possible range of values that the missing value could have taken.
>
> *Action:* We will explicitly clarify how our proposed blocking approach resolves missingness in Section 1 and introduce a new Figure 1 that visually depicts how our policy addresses retrospective missingness (please refer to the response for **[W2]** in Reviewer **xwXF**).
>
> We appreciate your time and consideration. If you have additional questions, please let us know. In light of our response, we hope you consider improving the score.
>
> [1] Nagler, T. (2023, July). Statistical foundations of prior-data fitted networks. In International Conference on Machine Learning (pp. 25660-25676). PMLR.

---

> > ### Author Rebuttal · Reviewer_huwr · 2026-04-03
> >
> > Thank you for your response.
> > The problem setting of AFA using meta-learning is interesting, and if the paper's presentation is improved as denoted in the authors' response, I will raise my rating. However, since I am not an expert in this area (AFA), it is difficult for met to make a fully confident assessment. Thus, I would ask the meta-reviewer to take this into consideration when making the final judgement.

---

> > > ### Author Response · Authors · 2026-04-05
> > >
> > > We sincerely thank the reviewer for their willingness to reconsider the rating. As edits to the draft cannot be made during the rebuttal period, we respectfully ask that the score be updated during this period based on the detailed revisions provided in our response, which we consider valuable feedback and will incorporate in the final manuscript.
> > >
> > > We provide the new changes made to the subsection and text to clearly delineate the test-time acquisition procedure under unobserved labels, and training with observed labels:
> > > > **4.1 Test-Time Feature Acquisition via Meta-Learned Sequence Models.** We consider a dataset $Z^{1:m}=\\{X^{1:m},R^{1:m},Y^{1:m}\\}$ of $m$ historical samples drawn from a task-specific observational distribution, along with $N-m$ query samples $\\{X_0^{m+1:N}\\}$ with partially observed features and unobserved targets from the same task distribution. The objective is to sequentially acquire features for each query sample in order to infer its unobserved target. Specifically, for each query instance $q \in {m+1,\dots,N}$ at acquisition step $t$, we select the next feature to acquire based on its currently observed features $\underline{X}_t^{(q)}$ and the task-specific context provided by the historical data $Z^{1:m}$...
> > >
> > > > **4.2 Meta-Learning the Predictor and Acquisition Policy.** Rather than computing the CMI exactly, we adopt the practical approximation of Covert et al. (2023, Prop. 2): we optimize the one-step-ahead predictive loss of a meta-predictor $f_\phi$ for $Y$ after acquiring a candidate feature $X_j$. A meta-policy $\pi_\theta$ is then trained to directly minimize this one-step loss, providing a tractable surrogate for the CMI objective. We now describe how both the predictor and the acquisition policy are meta-learned from training data...
> > >
> > > We additionally refer the reviewer to our response to Reviewer **6mve**, which details further revisions we are making to improve clarity.

---

### Official Review · Reviewer_xwXF · 2026-03-07

**Soundness:** 2
**Presentation:** 2
**Significance:** 3
**Originality:** 3
**Overall Recommendation:** 4
**Confidence:** 4

**Summary:**

This paper introduces Learning-to-Measure (L2M), a framework that scales Active Feature Acquisition (AFA) to the “Meta-AFA” setting. Unlike traditional methods that require training a bespoke policy for a single predetermined task, L2M learns a universal acquisition policy capable of generalizing across diverse tasks. It leverages autoregressive sequence modeling to perform reliable uncertainty quantification on datasets with retrospective missingness. During inference, L2M operates entirely in-context, using a greedy policy to sequentially acquire the most informative features that maximize conditional mutual information. This allows the model to adaptively select features for unseen tasks without any per-task retraining or fine-tuning.

**Compliance With Llm Reviewing Policy:**

Affirmed.

**Final Justification:**

The authors have successfully addressed my primary concerns. Consequently, I have raised my score from 3 to 4.

**Key Questions For Authors:**

Please look strength and weakness

**Limitations:**

yes

**Strengths And Weaknesses:**

Strength:

1.	The paper tackles a critical and widely acknowledged bottleneck in real-world AFA: retrospective missingness. Unlike many existing works that rely on the utopian assumption of fully observed training data, this paper directly confronts the messy reality of systematically missing features in historical datasets. This makes the research highly significant and practically valuable for downstream deployments.
2.	The introduction of a meta-learning and in-context learning framework for AFA provides a highly promising future direction for the field. Instead of the computationally expensive "train-from-scratch" paradigm required by traditional task-specific models, L2M demonstrates the ability to generalize across diverse tasks with minimal task-specific data. This zero-shot/few-shot capability significantly enhances the framework's scalability and flexibility in dynamic environments.
3.	This paper provides rigorous theoretical guarantees that address the unbiased estimation of CMI from incomplete historical datasets and surrogate optimization optimality in AFA.

Weakness:

1.	(Soundness) A primary claim of this paper is its ability to handle "retrospective missingness" in the training data natively. However, the experimental validation (Figure 3) relies solely on older baselines from 2023 (such as GDFS and DIME), which were originally designed under the assumption of fully observed training dataset. Compare with the baseline that is not targeted with same scope, retrospective missingness, is not convincing enough. To make the evaluation truly convincing, the authors should compare their approach against recent advancements that explicitly tackle historically missing data in AFA), such as [1].
2.	(Presentation) The teaser figure (Figure 1) is currently hard to follow and visually misleading, failing to effectively communicate the core "meta-learning" contribution of the L2M framework.

   a.	Although the paper presents L2M as a unified meta-learning approach, the visual layout depicts three separate pipelines for Tasks 1, 2, and 3. This makes it look like standard task-specific models are being trained independently.

   b.	The overall logical workflow (Input $\rightarrow$ Process $\rightarrow$ Output) is cluttered. The bottom label mentions "$b$ acquisition steps," but there is no visualization, and no sequential loop.

   c.	The use of red and blue font colors in the outputs is confusing. And fail to demonstrate the effectiveness  (no ground truth, and no improvement illustration)


Minor weakness:

1.	Reference of DIME need an update.
2.	In figure 3, pretrain part – step 1, what is M means?

[1] von Kleist, H., Zamanian, A., Shpitser, I., & Ahmidi, N. (2025). Evaluation of active feature acquisition methods for time-varying feature settings. Journal of Machine Learning Research, 26(60), 1-84.

---

> ### Author Rebuttal · Authors · 2026-03-31
>
> We thank the reviewer for their detailed and constructive feedback, and address the concerns below.
>
> **[W1] Evaluation of missingness-aware baselines.** We thank the reviewer for raising this point and appreciate the opportunity to clarify an important distinction. The aim of the referenced work [1] is not to design new AFA agents but to **evaluate** the performance of any AFA agent using retrospectively missing data during deployment. Specifically, it estimates the expected cumulative reward (value estimate) for an entire acquisition trajectory under a stochastic policy for the MDP. In contrast, L2M performs greedy policy **optimization**, where the argmax over actions reduces the problem to *learning a relative ordering of candidate features at each step*. This contrasts with their approach of estimating a full expectation over the policy’s induced distribution of feature acquisition trajectories. Therefore, neither [1] nor other prior work offers a directly comparable approach to our meta-AFA framework beyond the task-specific baselines.
>
> We will include a more explicit discussion of this distinction in the Appendix of the revised manuscript to clarify the relationship between our work and [1], and to better contextualize our choice of baselines. We also note that our baselines DIME/GDFS were modified to perform policy optimization under our missingness scenarios using our approach.
>
> **Additional Experiment:** We include a new experiment to clarify why our approach may be more desirable for policy learning from a practical standpoint. Since we cannot directly provide policy performance comparisons, we compare the quality of the policy gradient estimator.  We compute a reference policy gradient estimate from an unbiased REINFORCE estimator using the fully observed data over 5 actions. Then we compare the quality of the policy gradient estimate of our proposed surrogate pathwise estimators and the offline and semi-offline IPW estimators in [1] using partially observed data. We evaluate gradient estimate quality using three metrics relative to the full-data REINFORCE reference: (i) bias norm, measuring absolute estimation error, (ii) relative bias, normalizing error by the reference gradient magnitude, and (iii) cosine similarity, measuring directional alignment independent of magnitude.
>
> **Table 1.** Bias vs full-data REINFORCE 5-step reference (n=500 samples).
> Mean ± std across 50 sampled synthetic tasks. n denotes number of MC samples used for estimation. T denotes temperature parameter.
> | Estimator | Bias norm | Rel. bias | Cosine sim |
> |---|:--:|:--:|:--:|
> | Pathwise 1-step (T=0.5, n=4) | **0.068 ± 0.037** | **0.88 ± 0.33** | 0.58 ± 0.29 |
> | Pathwise 2-step (T=0.5, n=4) | 0.075 ± 0.049 | 0.95 ± 0.41 | **0.67 ± 0.26** |
> | Offline IPW REINFORCE 5-step (n=64) | 0.094 ± 0.022 | 1.56 ± 1.07 | 0.55 ± 0.32 |
> | Offline IPW REINFORCE 5-step (n=128) | 0.072 ± 0.022 | 1.16 ± 0.71 | 0.62 ± 0.31 |
> | Semi-offline IPW REINFORCE 5-step (n=64) | 1.215 ± 0.375 | 19.98 ± 14.29 | 0.29 ± 0.29 |
> | Semi-offline IPW REINFORCE 5-step (n=128) | 1.128 ± 0.345 | 18.70 ± 14.43 | 0.31 ± 0.29 |
>
> We use logistic regression models as in [1] to estimate the propensities $P(R=1 \mid X_0)$. We find that methods introduced in [1] can suffer from high variance and are susceptible to model approximation errors, especially for the semi-offline IPW approach which requires computing products over marginal inverse propensity weights over the trajectory. Our pathwise estimators are much more data efficient, only requiring 4 samples to perform comparably with the estimators from [1] that use n=64 or 128 samples.
>
> **[W2] Figure clarity.** We agree that Figure 1, as currently presented, only visualizes the problem setup of meta-AFA and does not effectively communicate our proposed contribution. In the revised manuscript, we will redesign Figure 1 by separating the problem setup from the meta-learning framework, explicitly illustrating how our approach unifies learning across tasks rather than training independent per-task models. We will also incorporate a clear depiction of the sequential acquisition process to convey the iterative nature of the framework. Finally,  we apologize for the confusion caused by the color scheme. Red and blue numbers denote the acquired feature value and final label prediction, respectively. We will revise the color scheme to improve readability, and concretely illustrate how our approach is effective under retrospective missingness.
>
> **[W3]** Thank you, we will update the citation for DIME.
>
> **[W4] Figure Notation.** We apologize for the confusion, this is a typo. The variable should be denoted R (missingness) rather than M. We will correct this in the revision.
>
> We appreciate your time and consideration. If you have additional questions, please let us know. In light of our response, we hope you consider improving the score.

---

> > ### Author Rebuttal · Reviewer_xwXF · 2026-04-01
> >
> > I thank the authors for their detailed response and the additional experiments/clarifications. The rebuttal has addressed my concerns, so I have increased my score.

---

### Official Review · Reviewer_6mve · 2026-03-08

**Soundness:** 3
**Presentation:** 2
**Significance:** 3
**Originality:** 3
**Overall Recommendation:** 4
**Confidence:** 2

**Summary:**

The paper proposes a new framework for Active feature acquisition to loosen the dependency on augmented retrospective data. The approach combines uncertainty quantification for unseen tasks and uses it for an uncertainty-guided feature acquisition, maximizing conditional mutual information. It further consists of two stages: a pre-training across multiple tasks for a task-agnostic method and the training of a policy network. The paper presents an extensive evaluation on synthetic and real-world tasks against other baselines.

**Compliance With Llm Reviewing Policy:**

Affirmed.

**Final Justification:**

The authors provided an extensive rebuttal on how they can improve the paper, which address the outlined weaknesses and improve the clarity and presentation of the results.
Given authors adapt their paper accordingly I increase my score.

**Key Questions For Authors:**

* If the Pretraining happens across tasks, how do training times differ from per-task approaches?
* What underlying model has been used? Does it repeat performance for other models, too?

**Limitations:**

yes

**Strengths And Weaknesses:**

## Strengths

* The paper provides a lot of experiments on different datasets, including real and synthetic data.
* The paper discusses the limitations of the method quite extensively.
* The proposed method addresses an important topic of task-agnostic feature acquisition, which seems to be effective.

## Major Weaknesses

The paper's structure and writing are partly confusing, making it hard to understand, and the results are presented with little discussion and context.

* The paper lacks a conclusion section. Instead, it has a short paragraph in the discussion that misses key results. Then it discusses the limitations quite extensively, but given the weaknesses in the experimental results presentation, the limitations feel disconnected.
* While the method section provides a good derivation of the equation, the exact implementation of the method should be more clearly explained and integrated with the Training section. Also, the transitions between the method's sections should better align with the high-level method. A full figure is presented in the experiments, which is way too late.
* The paper claims that L2M can also be used for LLM in the beginning for the methodology, but does not further discuss this in the experiments. In fact, the method is only tested on simple tasks.
* The Results are presented confusingly, the context between the figures changes rapidly, and the sentences reference the main paper and the appendix figure without distinction. This section should be structured to highlight the different aspects of the examined methods that are currently not well conveyed.
* The results discussion is quite short. The different metrics and tasks are hardly aligned with each other.
* A comparison of different runtimes is missing. Since the methods used BNN and across-task training, this method seems to have a high runtime.
* The Discussion, mixes

## Minor Weaknesses:

* Equations violate border margins.
* Some Typos: E.g. 375 punctuation missing

---

> ### Author Rebuttal · Authors · 2026-03-30
>
> We thank the reviewer for the careful reading, and constructive and detailed structural feedback. We will use the expanded page limit to significantly expand the results interpretation and conclusions, emphasizing key takeaways for readability.
>
> **Summary.** To summarize our contributions: we (i) propose the meta-AFA problem and establish assumptions for theoretical feasibility, (ii) develop an algorithm enabling in-context AFA, and (iii) empirically demonstrate reliable uncertainty estimation (Figures 4, 7–9) and acquisition behavior (Figures 3, 10) across in-distribution, out-of-distribution, synthetic, and real-world tasks compared to task-specific baselines.
>
> **[W1] Summary of key results and limitations.** We will expand the discussion, organized around the contributions summarized above, highlighting key results and their implications for each point. To improve clarity, we will reorganize limitations into conceptual and empirical categories, explicitly linking the latter to specific experimental results. For example, performance on unseen real-world data tasks (Figure 14) may be hindered by limitations in designing a diverse synthetic task prior (limitation ii).
>
> **[W2] Implementation details**  Our goal was to maintain a deliberate separation between the general conceptual framework for meta-learning acquisition policies and its specific implementation choices.
>
> *Action:* We will relocate Figure 2 to appear alongside the conceptual framework and improve transitions connecting the high-level methodology to its concrete realization. In particular, we will clarify the motivation behind key design choices: For example, permutation invariance in the transformer architecture is a desired property for modeling conditional distributions without sensitivity to context sample ordering.
>
> **[W3] Generality claims and scope of evaluation.** Please refer to the response for **[W2]** in Reviewer **8eNS**.
>
> **[W4] Results structure and readability.** We will restructure the section so that each subsection addresses a distinct comparison. For example, we compare meta-learning vs. task-specific, effect of lookahead steps, and greedy vs. full MDP optimization, with the key takeaway stated upfront in each case.
>
> *Action:* We will (i) restructure results around clearly delineated comparisons with explicit motivation for each, (ii) distinguish main-text figures from appendix references consistently, and (iii) provide additional context on comparator methods within each experiment.
>
> **[W5] Metrics and results analysis.** Our metrics are adopted in related works and are specifically chosen to rigorously validate the theoretical claims of our work. In particular, our metrics are specifically suited to evaluating uncertainty estimation, which is central to our framework. For instance, we employ log loss in Figure 3 to demonstrate that the acquisition policy exhibits the desired behavior induced by our log loss optimization objective, i.e., preferentially acquiring features that maximally reduce predictive uncertainty.
>
> **[W6] Compute details.** We provided compute details and approximate runtimes in Appendix A.7.4 and A.7.5. We will reference this section in the main text and provide more precise calculations of runtimes as shown below.
>
> (Ex.) On a single H100 GPU with synthetic BNN data (20 features, batch size 8, sequence length 1000), L2M processes 5000 training steps in **47 minutes** (0.57s/batch median).
>
> **[Q1] Training time comparisons.** Our method is designed to be pretrained on millions of synthetically generated tasks, analogous to the pretraining paradigm employed by foundation models such as TabPFN [1]. As such, the training times are not directly comparable to those of single-task models, as the two paradigms operate under fundamentally different assumptions: per-task approaches optimize for a single fixed-size dataset, whereas our approach learns across a diverse set of tasks with varying dimensionality and sample size. Large-scale pretraining has become a widely adopted paradigm, and we do not view this distinction as a limitation of our method but rather as a deliberate design choice.
>
> **[Q2] Architectural choices.** As described in Section 4.2 and Appendix A.6, our implementation employs a permutation-invariant transformer with full attention, consistent with related work. In Appendix A.18, we evaluate an alternative architecture using a single pooled context embedding, demonstrating that our framework is not dependent on a single architectural choice. However, the primary contributions of this work are conceptual and algorithmic (see summary); our model architecture is kept simple and serves as a proof of concept. A comprehensive evaluation across broader architectures (e.g., recurrent/state space models) falls beyond the scope.
>
> [1] Hollmann, N., Müller, S., Eggensperger, K., & Hutter, F. (2022). Tabpfn: A transformer that solves small tabular classification problems in a second.

---

> > ### Author Rebuttal · Reviewer_6mve · 2026-04-03
> >
> > [W1]
> > - I criticized the absence of a clear conclusion section. The authors answered that they will extend the discussion section. I do not see that the proposed approaches improved clarity.
> >
> > [W2]
> > - The proposed suggestion increases the clarity. However, how do the authors aim to improve the implementation details?
> >
> > [W3]
> > - Withdraws the unsupported claim, which corrects the paper but limits the scope. Thanks for correcting.
> >
> > [W4]
> > - Resolved.
> >
> > [W5]
> > - Ok, I got why you chose the given metrics. However, I am wondering about a detailed analysis of the results given in the figures. What is the interpretation of the results? This should be done together with W4. I consider it mostly resolved with W4.
> >
> > [W6]
> > - I ask for a runtime comparison. While the provided Appendix reference lists the run time, it does not compare it with other methods, which we also asked for in Q1. I understand that this might not be comparable, given the different setup, but it is part of a reflected evaluation of the method.
> >
> >
> > Could you provide more details on the partly addressed weaknesses?

---

> > > ### Author Response · Authors · 2026-04-05
> > >
> > > Thank you for the additional questions.
> > >
> > > **[W1]** We will replace the first paragraph of what is currently titled “Discussion” with a dedicated conclusion section:
> > >
> > > > **Conclusion**: In this work, we introduced the meta-AFA problem and presented L2M, an end-to-end differentiable, uncertainty-driven framework for feature acquisition that performs in-context learning across tasks. Our contributions are threefold. First, we formalized the meta-AFA problem and established the theoretical assumptions under which meta-learning acquisition policies is feasible. Second, we developed an algorithm that enables in-context active feature acquisition, leveraging meta-learned approximations of target conditional distributions for AFA policy optimization. Third, we empirically validated our framework across synthetic and real-world settings, demonstrating reliable uncertainty estimation and principled acquisition behavior under both in-distribution and out-of-distribution conditions compared to task-specific baselines. Notably, L2M exhibits robust performance under limited labeled data and significant retrospective missingness, conditions that pose substantial challenges for existing approaches. We believe that the meta-AFA formulation opens promising avenues for future research, including...
> > >
> > > **[W2]** We will restructure Section 4.2 by replacing the final two paragraphs with a dedicated subsection on model architecture and training with additional details. Below is the revised model architecture description as an illustration:
> > >
> > > > **Model Architecture**:  While our conceptual framework and policy optimization algorithm are agnostic to the specific parameterization of sequence models, we propose a joint Transformer encoder with a separate target predictor and acquisition policy head as a proof of concept. As summarized in Figure 2, the Transformer takes as input the query sample and the observational dataset, represented as a sequence of samples with missingness, and produces encodings that are passed to the target predictor and acquisition policy heads. Relative to a standard Transformer, we make the following key changes to model the posterior predictive distributions in Equation 4. (i) To represent missing features, we represent each sample $i$ in a sequence as the concatenation of its masked features, missingness indicator, and label: $[\mathbf{x}^i \odot \mathbf{r}^i, \mathbf{r}^i, y^i]$. The use of masked features alongside a missingness indicator is a common strategy in prior AFA work (Covert et al., 2023; Gadgil et al., 2023; Norcliffe et al., 2025). (ii) To make the model invariant to the ordering of samples in the sequence, we remove positional embeddings and replace causal masking with an alternative attention masking structure during both training and inference for permutation invariance, consistent with related work (Nguyen et al., 2022; Ye et al., 2024).
> > >
> > > **[W5]** We will expand the results discussion with additional interpretation in conjunction with the structural changes outlined in [W4]. To illustrate the level of detail we intend to provide, we include below a revised version of the first paragraph of the results section:
> > >
> > > > **L2M improves uncertainty quantification over task-specific models.** We first demonstrate that meta-learning across diverse tasks improves uncertainty quantification relative to models trained independently on each task. We evaluate uncertainty quality using three complementary metrics: empirical vs. nominal coverage (Kompa et al., 2021), log loss, and mean squared error (MSE) (Nguyen et al., 2022). Figure 3 shows that L2M outperforms task-specific MLP baselines on coverage in synthetic GP regression tasks. Additional analysis in Appendix Figure 8 confirms that L2M also achieves lower log loss and MSE. Further results on semi-synthetic and real-world classification tasks are reported in Appendix Figures 9 and 10. A key finding is that in the presence of missingness in training, L2M provides *increasingly reliable uncertainty estimates as more features are acquired*, whereas task-specific MLPs degrade at later acquisition steps due to reduced joint feature coverage caused by retrospective missingness. We attribute this advantage to L2M leveraging its diverse learned task prior to mitigate the effects of reduced coverage. This improvement in uncertainty quantification is particularly significant, as it directly translates to stronger downstream acquisition performance for both the RBF and Matérn kernel tasks, as shown in Figure 5.
> > >
> > >
> > > **[W6]** We will include the precise training times for our baseline methods using the same synthetic BNN data (Number of features (10-20) and sample size (50-1000) varies per task):
> > > > Baseline classifier train time (seconds): n=50 tasks, 300 epoch, median=11.65, total=599.85
> > >
> > > > GDFS train time (seconds): n=50 tasks, 50 epoch, median=52.91, total=2549.27
> > >
> > > > DIME train time (seconds): n=50 tasks, 50 epoch, median=37.89, total=1804.40

---

### Official Review · Reviewer_8eNS · 2026-03-10

**Soundness:** 4
**Presentation:** 3
**Significance:** 3
**Originality:** 3
**Overall Recommendation:** 4
**Confidence:** 4

**Summary:**

Active feature acquisition allows a model to select the features to acquire at inference. The idea is that there are missing features or corrupted features in various contexts that should be avoided for accurate test time predictions. The authors propose learning to measure that performs reliable uncertainty quantification and selects features with the idea of maximizing mutual information.

**Compliance With Llm Reviewing Policy:**

Affirmed.

**Final Justification:**

The proposed action of adjusting the generality of the claims in the final version is also good. I believe the paper could have been much stronger if the authors had showcased this generality experimentally. However, the paper has multiple merits and I will keep my score.

**Key Questions For Authors:**

1. How do you think more complex feature dependencies on larger scale datasets would impact the performance and trainability of your method?

2. What alternate considerations exist when leveraging your method on a larger model such as a pre-trained LLM or in other settings beyond tabular data?

**Limitations:**

Yes

**Strengths And Weaknesses:**

# Strengths:

The authors do a good job of justifying the setting of their problem. There are many situations where previous AFA approaches are too expensive, biased, or infeasible to deploy in real world settings. This motivation leads to the ideas of their algorithm. Their approach makes a lot of sense from this motivation by combining both uncertainty estimation with a sequential modeling process to learn a policy for learning the best features. The authors then demonstrate theoretically how their proposed approach corresponds to choosing features that maximize conditional mutual information. These insights are discussed carefully in order to bring intuition to all aspects of the work.

I also feel that this work exhibits originality in the sense that the reinforcement policy alongside uncertainty estimation framework is a principled approach to solving this problem.

The authors perform relevant empirical studies and comparisons. I especially like that they compare their feature selection strategy against relevant baselines such as the DQN network.

# Weaknesses:

I enjoyed reading this paper, but I think most of the weaknesses stem from the scale of the datasets that were used in the evaluation of the given method. For example, MIMIC, MNIST, and metabric can be considered small scale datasets. I believe that this problem is especially relevant when one considers the nature of the problem. For example, the authors rely on a policy network to learn features to select. However, in general, it is difficult to train policy networks to converge to good solutions. It may be the case that on larger datasets with more complex feature dependencies it might be intractable to use this method to learn a good feature selection strategy.

Furthermore, the authors write as if this is a general strategy, but the only datasets used for validation are tabular datasets and MNIST. At different points in the paper, the authors discuss other potential settings for their method such as usable on a pre-trained large model or in other contexts beyond tabular data. The generality argument of the method would be better justified if some further ablation studies could be examined, such as analyzing your method within the context of fine-tuning.

Additionally, the missing feature problem is prevalent in many other contexts beyond just tabular data. This could include contexts such as missing sensor signals or redacted portion of NLP documents. It would have been good to have one or two more of these settings to really solidify the claims of this paper.

---

> ### Author Rebuttal · Authors · 2026-03-29
>
> We thank the reviewer for their positive assessment and thoughtful questions, which give us the opportunity to clarify the scope of our work.
>
>
> **[W1, Q1] Scaling and complex feature dependencies.**
> We agree that scaling is an important engineering direction and acknowledge this limitation. Validating at larger scales requires significant compute resources that are beyond our current capacity. Our main contributions are (i) conceptualizing meta-AFA and establishing assumptions for theoretical feasibility, and (ii) providing empirical evidence on multiple datasets to carefully validate model behavior in small-scale settings.
>
>
> We clarify that complex feature dependencies and larger feature counts are precisely why we opt to meta-learn greedy/one-step lookahead policies as a practical alternative to optimizing the full MDP. Full-horizon policy networks (e.g., DQN, PPO), while theoretically optimal, are widely known to be difficult to train even for a single dataset. Past AFA work and benchmarks, including single-task scenarios, have been mostly evaluated at small scales due to this difficulty [1].
>
>
> We also note that complex feature dependencies and larger feature spaces present a recurring practical challenge for all feature selection and acquisition methods, not only ours. For example, static feature importance approaches such as SHAP rely on marginalizing over absent features, which in standard implementations assumes feature independence for tractability rather than explicitly modeling dependencies. To our knowledge, Figure 5 provides the first empirical evidence that feature interactions can be captured by meta-learning across a diverse distribution of synthetic tasks with varying dependency structures.
>
>
>
> **[W2, Q2] Generality claims and scope of evaluation.**
> We apologize for the confusion. Our method is designed specifically for structured data. The generality of our conceptual framework stems from requiring only (i) approximations of one-step conditional distributions and (ii) differentiability. This makes our conceptual contribution compatible with other input representations and sequence modeling architectures (e.g., pretrained LLMs repurposed for tabular classification via text serialization [2]). However, token-based inputs cannot be smoothened like masking, and may require different approximations to bypass the lack of differentiability.
>
> *Action:* As using pretrained LLM encoders was not empirically validated, we recognize the claim is insufficiently supported. We will remove this statement and clearly delineate the scope of our experimental evaluation.
>
> **[W3, Q2] Time-varying features and sensor data.**
> Our theoretical framework and proposed method are specifically designed for the time-invariant setting, wherein feature values are static and do not evolve over time. Settings such as sensor data or time-series fall outside this scope, as they introduce the need to model temporal dynamics governing how feature values evolve. This is a substantively different problem requiring distinct theoretical treatment.
>
> *Action:* While we note this scope boundary in limitation (v), we will add explicit examples and discussion of these additional considerations.
>
> **References**
>
> [1] Schütz et al. AFABench: A Generic Framework for Benchmarking Active Feature Acquisition. arXiv 2508.14734, 2025.
>
> [2] Gardner et al. Large Scale Transfer Learning for Tabular Data via Language Modeling. NeurIPS 2024.

---

> > ### Author Rebuttal · Reviewer_8eNS · 2026-04-04
> >
> > I thank the authors for the thoughtful rebuttal and have gone through other reviews as well.
> >
> > I understand the practical challenges in computation. The proposed action of adjusting the generality of the claims in the final version is also good. I believe the paper could have been much stronger if the authors had showcased this generality experimentally. However, the paper has multiple merits and I will keep my score.

---

### Decision · Program_Chairs · 2026-04-30

**Decision:**

Accept (regular)

**Comment:**

This paper introduces Learning-to-Measure (L2M), a meta-learning framework for in-context active feature acquisition under retrospective missingness. Reviewers agreed that the paper addresses an important and practically relevant limitation of prior AFA methods, namely the dependence on fully observed settings, and they found the combination of uncertainty estimation and acquisition policy learning technically meaningful. The rebuttal was generally effective in clarifying the scope of the method, and addressing concerns about evaluation and presentation, leading reviewers to maintain or raise their scores. The main remaining weaknesses concern clarity of presentation, and the need for clearer discussion of runtime and implementation details. However, these concerns appear manageable in revision and do not outweigh the paper’s conceptual novelty and empirical promise. Overall, I believe this is a worthwhile contribution to the area. I recommend acceptance.